# Controllable CO$_2$ electrocatalytic reduction via ferroelectric switching on single atom anchored In$_2$Se$_3$ monolayer

Lin Ju[1,2,9], Xin Tan[3,9], Xin Mao[4], Yuantong Gu [1,5,6], Sean Smith[3], Aijun Du[4,5], Zhongfang Chen [7], Changfeng Chen [8] & Liangzhi Kou [1,5✉]

Efficient and selective CO$_2$ electroreduction into chemical fuels promises to alleviate environmental pollution and energy crisis, but it relies on catalysts with controllable product selectivity and reaction path. Here, by means of first-principles calculations, we identify six ferroelectric catalysts comprising transition-metal atoms anchored on In$_2$Se$_3$ monolayer, whose catalytic performance can be controlled by ferroelectric switching based on adjusted *d*-band center and occupation of supported metal atoms. The polarization dependent activation allows effective control of the limiting potential of CO$_2$ reduction on TM@In$_2$Se$_3$ (TM = Ni, Pd, Rh, Nb, and Re) as well as the reaction paths and final products on Nb@In$_2$Se$_3$ and Re@In$_2$Se$_3$. Interestingly, the ferroelectric switching can even reactivate the stuck catalytic CO$_2$ reduction on Zr@In$_2$Se$_3$. The fairly low limiting potential and the unique ferroelectric controllable CO$_2$ catalytic performance on atomically dispersed transition-metals on In$_2$Se$_3$ clearly distinguish them from traditional single atom catalysts, and open an avenue toward improving catalytic activity and selectivity for efficient and controllable electrochemical CO$_2$ reduction reaction.

[1] School of Mechanical, Medical and Process Engineering Faculty, Queensland University of Technology, Brisbane, QLD, Australia. [2] School of Physics and Electric Engineering, Anyang Normal University, Anyang, China. [3] Integrated Materials Design Laboratory, Department of Applied Mathematics, Research School of Physics, The Australian National University, Canberra, Australian Captial Territory, Australia. [4] School of Chemistry and Physics, Queensland University of Technology, Brisbane, QLD, Australia. [5] Center for Materials Science, Queensland University of Technology, Brisbane, QLD, Australia. [6] Centre for Biomedical Technologies, Queensland University of Technology, Brisbane, QLD, Australia. [7] Department of Chemistry, University of Puerto Rico, Rio Piedras Campus, San Juan, PR, USA. [8] Department of Physics and Astronomy, University of Nevada, Las Vegas, NV, USA. [9] These authors contributed equally: Lin Ju and Xin Tan. ✉email: liangzhi.kou@qut.edu.au

The ever-rising global energy consumption and its negative impact on the environment are major challenges in front of humanbeing in the 21$^{st}$ century[1–7]. Public health is severely threatened by the greenhouse effect due to the excessive fossil fuel usage. It is highly desirable to develop technologies that can convert the greenhouse gas (i.e., $CO_2$) into sustainable and clean energy sources. Among various possibilities, catalytic reduction of $CO_2$ into hydrocarbon fuels, which features environmental friendliness, high efficiency, and low cost, has been recognized as one of the most promising approaches. Single atom catalysts (SACs), first proposed for CO oxidation in 2011[8], offer us excellent candidates to activate and convert $CO_2$ efficiently when trasition metal (TM) atoms are used[9–13], because the coexistence of empty and occupied TM $d$-orbitals can accept lone-pair electrons, and then back-donate these electrons to the antibonding orbitals to weaken the C=O bonds. Notably, two-dimensional (2D) materials serve as promising substrates for atomically dispersed transition metal atoms for $CO_2$ reduction due to their large surface-to-volume ratios, short carrier diffusion distances, unique electronic properties, and abundant active sites. The catalytic performance of these 2D SACs stems from versatile advantageous characteristics. For example, catalysts with singly dispersed Ni, Co, Cu, Pt, and Pd atoms on graphene[14], Pt or Pd atom anchored on $C_3N_4$[15], $V$-$\beta_{12}$ boron monolayer[16] and Co decorated metal–organic frameworks[17,18] all have shown great potential for $CO_2$ reduction.

Despite latest advances, there is still plenty of room to improve the catalytic activity and selectivity of $CO_2$ reduction in order to satisfy large-scale industrial applications. Recent studies have shown that electric polarization plays an important role in determining catalytic activities and selectivities, and can greatly improve the catalytic performance compared with the unpolarized counterparts[19–21]. For instance, Janus TM dichalcogenides possess better catalytic abilities for water splitting and $N_2$ fixation[19–21], since the catalysis-related properties, including surface stoichiometry, electronic structure[22–26], adsorption strengths, and reaction activation energies, can be tuned by the polarization[27]; Moreover, by utilizing the polarization from the substrate, the overpotential of oxygen evolution reaction on $TiO_2$ surface can be strongly reduced[28].

Inspired by these recent achivements, we expect that ferroelectrics, $viz.$, materials with switchable ferroelectric polarization, hold promise as efficient catalysts since their unique reversible polarization provides an additional mechanism to adjust empty and occupied $d$-orbitals of adsorbed metal atoms. The activation of reactants can thus be controlled, while the catalytic properties of the supported metal, such as activity[29–32] and selectivity, can be tuned[33]. These expectations have been partially confirmed in $ABO_3$ perovskite ferroelectrics. For instance, reversing the polarization of ferroelectric $PbTiO_3$ substrate makes $CrO_2$ monolayer overcome the limiting factor stemming from the Sabatier principle, and thus display excellent catalytic behaviours for both $NO_x$ decomposition and CO oxidation[34]. Reorienting the polarization direction of the ferroelectric $PbTiO_3$ substrate can dramatically change the chemisorption energies of CO, O, C, and N on the $PbTiO_3$-supported Pt films, and alter the reaction paths of the dissociations[29]. The oxygen evolution reaction activity of $TiO_2$ film on a ferroelectric $SrTiO_3$ substrate can be strongly enhanced relative to unsupported $TiO_2$ due to the presence of dynamic dipoles in response to the charge on the adsorbed species, while the reaction path can be modulated via ferroelectric switching[28]. However, these predictions have not been experimentally realized so far, mainly due to the instability and depolarization in traditional $ABO_3$ perovskite ferroelectrics when the thickness is smaller than a critical value. Encouragingly, the emergence of 2D ferroelectric materials, exemplified by $In_2Se_3$[35],

$CuInP_2S_6$[36], and SnTe[37], provides an excellent opportunity to verify controllable catalysis in ferroelectrics. The realization of ferroelectricity at room temperature makes it feasible for 2D ferroelectrics to be used for controllable catalytic $CO_2$ reduction.

In this work, based on the experimentally available 2D ferroelectrics, we theoretically investigate the potential of using transition metal decorated $\alpha$-$In_2Se_3$ monolayers as the ferroelectric SACs toward electrochemical $CO_2$ reduction. Our efforts identify six catalysts with singly dispersed TM atoms anchored on the 2D ferroelectric substrate, namely $\alpha$-$In_2Se_3$ monolayer, with polarization downwards or upwards (denoted as TM@P↓-$In_2Se_3$ or TM@P↑-$In_2Se_3$). The switchable polarization can not only alter the reaction barrier and paths of $CO_2$ reduction, but also lead to different final products. It can even reactivate the stuck $CO_2$ reduction. These performance improvements stem from the synergistic effects of adjusted empty and occupied $d$-orbitals ($d$ orbital center) of adsorbed metal atom, polarization dependent electron transfer, and $CO_2$ adsorption energies under ferroelectric switching. These ferroelectric SACs, catalytic mechanism, and the polarization dependent catalytic $CO_2$ reduction open an avenue for controllable $CO_2$ reduction reaction ($CO_2RR$), and introduce a feasible approach to significantly improve the efficiency of such reactions.

## Results

**Screening ferroelectric catalysts.** The ferroelectric $\alpha$-$In_2Se_3$ monolayer was chosen as the substrate of potential catalysts due to the following reasons: (1) it is the ground-state phase among possible phases (see Supplementary Fig. 1), even in the presence of adatoms and in electrochemical environment (see Supplementary Fig. 2 and Supplementary Tables 1–2), which is consistent with the recent theoretical finding by Ding et al.[38] Note that two $\alpha$-$In_2Se_3$ ($\alpha$ and $\alpha'$) phases have been predicted to share similar atomic structures and are almost energetically degenerate, and they have the same $CO_2RR$ performance (Supplementary Fig. 3). We thus mainly focus on $\alpha$-$In_2Se_3$ in this study. (2) it is stable at room temperature and has been experimentally synthesized[35,39], and its ferroelectricity with switchable polarization under room temperature has been explicitly demonstrated[35,40]. With the polarization locking from the asymmetric arrangement of its quintuple layers (see Fig. 1a, b), their orientations can be reversed by the shift of the middle Se layer[38]. However, as a ferroelectric semiconductor with a sizable band gap (1.46 eV)[38], $\alpha$-$In_2Se_3$ monolayer does not have enough electrons to be injected into the antibonding $2\pi_u$ orbitals of $CO_2$ so that the strong $sp$-hybridization symmetry of the carbon atom can not be disrupted[41]. Thus, the material itself is not suitable as a catalyst for $CO_2$ reduction[42], which is corroborated by our theoretical study: upon adsorption on $In_2Se_3$, the inherent linear O=C=O structure of $CO_2$ molecule is well maintained (see Supplementary Fig. 4 and Supplementary Table 3).

To activate electrochemical $CO_2$ reduction, we introduced transition metal atoms to decorate $\alpha$-$In_2Se_3$ monolayer, which provide extra electrons to break the strong $sp$ hybridization, thus activating the $CO_2$ molecules[29,42–44]. Compared with conventional metal catalysts, SACs normally exhibit higher catalytic performances because of the high ratio of low-coordinated configurations[8,45,46]. However, the transition metals have to be screened since only the SACs with well balanced empty/occupied $d$ orbitals can exhibit optimal catalytic performance. To find suitable SACs, 29 transition metal atoms were chosen and chemically adsorbed (see Supplementary Fig. 5) on both surfaces of the $In_2Se_3$ monolayer, and the energetically most favorable configurations were identified. The possible metal substitutions into the $In_2Se_3$ monolayer were excluded due to the large formation energy of In vacancy, high diffusion barriers of In

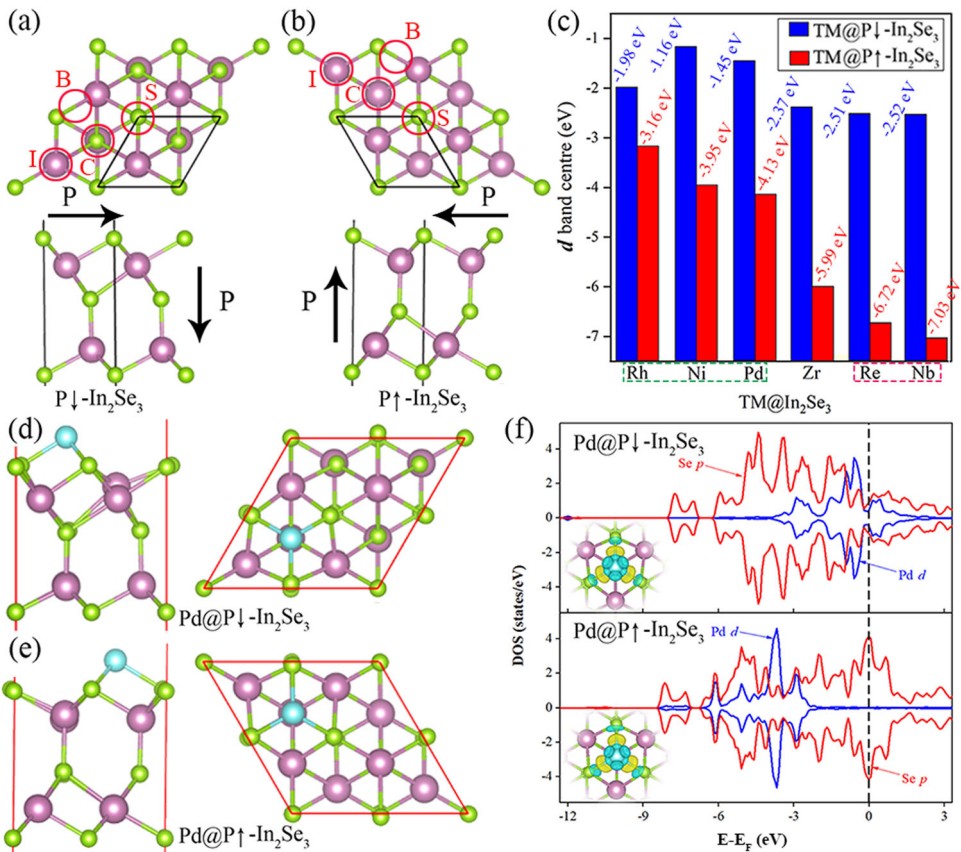

**Fig. 1 Geometries and electronic structures of the ferroelectric catalysts.** Top and side views of the optimized α-In$_2$Se$_3$ monolayer with **a** downward (P ↓ ) and **b** upward (P ↑ ) polarization. Red circles denote selected adsorption sites: I, top of the In atom; C, center of the six-membered ring; B, top of the In-Se bond; S, top of the Se atom. The black rhombus represent the unitcell of In$_2$Se$_3$. **c** The $d$ band center of TM@In$_2$Se$_3$ (TM = Ni, Pd, Rh, Zr, Nb and Re). Top and side views of the optimized configuration of **d** Pd@P ↓ -2×2 In$_2$Se$_3$ and **e** Pd@P ↑ -2×2 In$_2$Se$_3$, the supercell is indicated by the red rhombus. **f** The partial density of states of Pd@P ↓ -In$_2$Se$_3$ and Pd@P ↑ -In$_2$Se$_3$. The insets show the 3D differential charge density plots obtained with the model shown in Fig. 1d and e. The isosurfaces are 0.005 e/Å$^3$. Charge accumulation and depletion are marked by the yellow and blue regions, respectively.

atoms (see Supplementary Fig. 6), and low CO$_2$RR activities (see Supplementary Fig. 7). Then we evaluated all the composites based on two key criteria. First, to ensure the stability of the active site, the single transition metal atom should be steadily adsorbed on the In$_2$Se$_3$ monolayer surface without breaking the underlying structure, and the favourable adsorption site should not change significantly after the polarization switching; second, the single transition metal atom should be able to activate CO$_2$, in other words, the linear structure of O=C=O should be broken after adsorption, at least in one polarization state.

After a comprehensive investigation of the energetically most favorable adsorption configurations (Fig. 1a, b) and their stabilities, we screened out six TM atoms (TM = Ni, Nb, Pd, Re, Rh and Zr) as promising catalysts toward CO$_2$ reduction for further examinations. Other 23 candidates are either unstable or cannot effectively activate CO$_2$ molecules (see details in the *Supplementary Information*).

**Stability of SACs and polarization dependent $d$ band center.** Although the screened TM@In$_2$Se$_3$ are stable (without structural breaking) and able to activate CO$_2$ molecules, transition metal atoms need to uniformly disperse on the substrate to prevent any aggregation of the deposited atoms and maximize the catalytic efficiency[47].

To achieve this goal, strong binding strength, high diffusion energy barrier, and negative clustering energy are

essential[15,48–54]. At the energetically most favorable site, all the six TM atoms have rather strong binding energies (Table 1) that are comparable to previously predicated SACs[15,49,55,56], indicating sufficiently strong interactions between the metal and the substrate, as evidenced by the strong TM–Se bonds (Fig. 1d, e and Supplementary Fig. 8). To provide the reference for the binding strengths, we compared the binding energies of TM atoms in ferroelectric SACs with the ones in the corresponding molecular precursors, it is found that the adsorption of TM (e.g. Pd, Rh and Ni) atoms on In$_2$Se$_3$ is energetically preferred, indicating the feasibility of the SACs to be synthesized based on the corresponding molecular precursors (see Supplementary Table 4). The chemical bonding interaction is illustrated by the substantial partial charge accumulation between the TM atoms and the surrounding Se atoms (see Table 1, and Fig. 1f and Supplementary Fig. 8) as well as the strong hybridization of Se $p$ and TM $d$ orbitals (shown in Fig. 1f and Supplementary Fig. 9) that is similar to the case of TM atoms adsorbed on 2D carbo-nitride[15,49,55]. Moreover, the high diffusion barriers of TM atoms on the surfaces can effectively prevent the formation of metal clusters (see Supplementary Fig. 10), though specific values depend on the ferroelectric polarization.

To further demonstrate the structural stability of the SACs chosen in this work, we also conducted clustering energy calculations and *ab initio* molecular dynamics (AIMD) simulations with a Nose–Hoover thermostat at 300 K on Pd@In$_2$Se$_3$ as a representative example. The neagive clustering energies (−0.3 eV

**Table 1 Parameters for the pure TM@In$_2$Se$_3$ and CO$_2$ adsorbed TM@In$_2$Se$_3$: adsorption site ($S_{ad}$), binding energies of TM atom ($E_{b\text{-}TM}$ in eV/atom) and CO$_2$ molecule ($E_{b\text{-}CO2}$ in eV/molecule), charge lost from the adsorbed TM atoms ($Q_{TM}$ in $e$/atom), average TM-Se bond length ($l_{TM\text{-}Se}$ in Å), migration barrier ($E_{ba\text{-}TM}$ in eV), charge gained by the adsorbed CO$_2$ molecule ($Q_{CO2}$ in $e$/molecule), and bond angle ($\angle$OCO in °).**

| Catalysts | $S_{ad}$ | $E_{b\text{-}TM}$ | $E_{ba\text{-}TM}$ | $Q_{TM}$ | $l_{TM\text{-}Se}$ | $E_{b\text{-}CO2}$ | $Q_{CO2}$ | $\angle$OCO |
|---|---|---|---|---|---|---|---|---|
| Ni@P ↓ -In$_2$Se$_3$ | C | −2.72 | 1.78 | 0.40 | 2.31 | −1.39 | 0.45 | 146.4 |
| Ni@P ↑ -In$_2$Se$_3$ | C | −1.97 | 1.41 | 0.37 | 2.37 | −1.08 | 0.40 | 146.5 |
| Pd@P ↓ -In$_2$Se$_3$ | C | −1.69 | 1.19 | 0.14 | 2.50 | −0.84 | 0.33 | 149.2 |
| Pd@P ↑ -In$_2$Se$_3$ | C | −1.14 | 0.84 | 0.12 | 2.58 | −0.79 | 0.28 | 151.2 |
| Rh@P ↓ -In$_2$Se$_3$ | I | −4.09 | 4.19 | 0.12 | 2.38 | −1.05 | 0.40 | 146.5 |
| Rh@P ↑ -In$_2$Se$_3$ | I | −2.36 | 2.49 | 0.08 | 2.49 | −0.76 | 0.26 | 153.9 |
| Zr@P ↓ -In$_2$Se$_3$ | I | −6.26 | 6.13 | 2.10 | 2.58 | −0.01 | 0.02 | 179.2 |
| Zr@P ↑ -In$_2$Se$_3$ | I | −5.01 | 5.06 | 2.03 | 2.61 | −3.39 | 1.13 | 126.2 |
| Nb@P ↓ -In$_2$Se$_3$ | I | −7.08 | 4.63 | 1.70 | 2.46 | −2.54 | 0.92 | 131.1 |
| Nb@P ↑ -In$_2$Se$_3$ | I | −4.73 | 4.21 | 1.61 | 2.59 | −2.47 | 0.85 | 134.7 |
| Re@P ↓ -In$_2$Se$_3$ | C | −2.36 | 5.18 | 0.90 | 2.34 | −2.11 | 0.69 | 134.8 |
| Re@P ↑ -In$_2$Se$_3$ | C | −1.07 | 3.25 | 0.71 | 2.36 | −2.02 | 0.61 | 136.8 |

and −0.04 eV for Pd@P ↓ -In$_2$Se$_3$ and Pd@P ↑ -In$_2$Se$_3$, respectively; see details in SI) indicate that it is not energetically favorable to form clusters on the catalyst surface. In the AIMD simulations, the out-of-plane polarization of ferroelectric In$_2$Se$_3$ monolayer can be well retained, and the metal atom stays anchored at the energetically favorable site even at room-temperature (300 K) for at least 15 ps (see Supplementary Fig. 11a). Besides, AIMD simulations at 300 K are also performed to investigate two dispersedly adsorbed Pd atoms on P ↓ -In$_2$Se$_3$ (2 Pd@P ↓ -In$_2$Se$_3$). The two Pd atoms can maintain dispersedly adsorbed features without metal clustering and structural phase transition for 15 ps (see Supplementary Fig. 11b). The distance between the two Pd atoms stays almost unchanged (as shown in the inset of Supplementary Fig. 11). The possible metal (e.g. Pd and Nb) agglomerations are excluded by the further kinetic Monte Carlo (kMC) simulations, the clusters will not form at the surface for 100 seconds at the room-temperature (see Supplementary Figs. 12–14). Overall, the strong adsorption energies, high diffusion barriers, negative clustering energies, molecular dynamics simulation, and kMC results all indicate strong stability of the 2D ferroelectric SACs.

Good stabilities of the ferroelectric substrate and the catalysts themselves are key prerequisites to achieve the goal of controllable catalysis. The stability of ferroelectric α-In$_2$Se$_3$ monolayer has been unambiguously demonstrated by our simulations (Supplementary Fig. 1) and also by the recent theoretical and experimental studies[35,36,39,40], but stabilities under harsh evironments for CO$_x$RR still need to be addressed. To this end, based on the dissolution potential[57–59], we have evaluated and confirmed the electrochemical stability of our proposed system under acidic conditions (see Supplementary Table 2).

Different from traditional SACs like Pd@C$_3$N$_4$, the switchable polarization in ferroelectric catalysts proposed herein provides an extra degree of freedom to modulate and control catalytic reactivities. The average TM-Se bond length, binding energy, and electron transfer are highly polarization dependent (Table 1). Compared with the corresponding catalysts with upwards polarization, TM@P ↓ -In$_2$Se$_3$ generally have shorter average TM-Se bond length, larger absolute value of binding energy, and more electron transfer, due to the large electrostatic potential difference on In$_2$Se$_3$ as in metal porphyrazine molecules[60]. The balance between empty and occupied $d$ orbitals, caused by the polarization dependent electron transfer, significantly affect the catalytic activity of TM@In$_2$Se$_3$. Taking Pd@In$_2$Se$_3$ as an example, the charge is redistributed between the Pd and Se atoms (see

Fig. 1f), and the $d$ orbitals of Pd atom is depleted. Compared with Pd@P ↑ -In$_2$Se$_3$, more electron transfer occurs in Pd@P ↓ -In$_2$Se$_3$, shifting the Pd-$d$ orbitals to higher energy closer to the Fermi level. As illustrated in Fig. 1c, the $d$ band center of Pd@P ↓ -In$_2$Se$_3$(−1.45 eV) is higher than that of Pd@P ↑ -In$_2$Se$_3$ (−4.13 eV), indicating that Pd@P ↓ -In$_2$Se$_3$ possesses better catalytic ability[46]. Similar phenomena occur in the other five TM@In$_2$Se$_3$ catalysts (see Fig. 1c) under the same mechanism: the $d$ band center of the TM atom is closer to the Fermi level when it is placed on P ↓ -In$_2$Se$_3$. Note that the position differences of the $d$ band center under opposite polarizations for TM@In$_2$Se$_3$ (TM = Nb, Re) are rather pronounced, while such differences for TM@In$_2$Se$_3$ (TM = Ni, Pd and Rh) are only moderate. These contrasting results imply that both reaction paths and barriers are affected by the polarization (see a detailed catalytic classification below). Considering that the localized electronic states near the Fermi level lead to high N$_2$ catalytic activity in the catalyst Co$_2$@GDY (cobalt dimer-anchored graphdiyne)[49], we expect that the modulation of the electronic structure of TM@In$_2$Se$_3$ caused by the polarization switching has a major influence on the subsequent CO$_2$ hydrogenation process.

Since the catalytic activity of the TM@In$_2$Se$_3$ systems is polarization dependent, the effective ferroelectric switching of TM@In$_2$Se$_3$ is essential to achieve controllable catalytic reduction. We reverse the polarization orientation of Pd@In$_2$Se$_3$ (as a typical example, see Supplementary Figs. 15 and 16) via a three-step concerted mechanism on the most effective kinetic path[38]. We find that the energy barrier of the polarization switching after Pd decoration is 0.29 eV/UC (0.15 eV/UC) for P ↑ to P ↓ (P ↓ to P ↑), which is slightly larger than the value for the bare ferroelectric layer (0.13 eV/UC), but it is still accessible to experimental manipulation[38]. It is noted that the ferroelecitrc switching behaviors are not significantly affected by the metal adsorption. Although the local area of metal anoching site is metallized due to the hybridization between the metal and surrounding Se atoms, the semiconduting feature of the whole substrate and the associated ferroelectricity can be well preserved (see Supplementary Fig. 17). Considering that the out-of-plane ferroelectric switching of thin In$_2$Se$_3$ flakes has been experimentally achieved[40], it should be feasible to achive similar polarization orientation reversal in TM@In$_2$Se$_3$ systems.

**Tunable activation of CO$_2$ by polarization switching.** Effective activation of CO$_2$ molecule is the key for the subsequent reduction. Based on the structural analysis, almost all the TM@In$_2$Se$_3$ catalysts could effectively activate CO$_2$ molecule (see Fig. 2 and

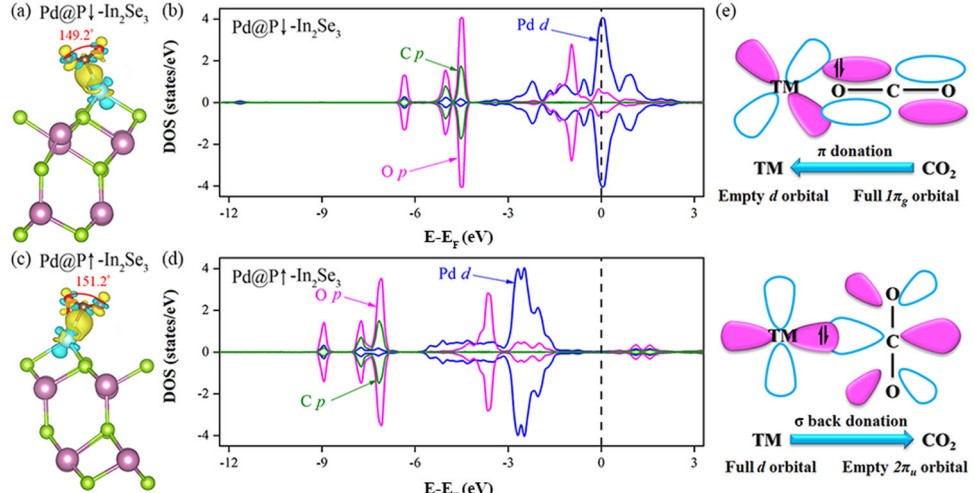

**Fig. 2 Tunable activation of CO₂ by polarization switching.** The differential charge density plots and partial density of states of CO₂ adsorbed on Pd@In₂Se₃ (**a, b** for Pd@P ↓ -In₂Se₃, and **c, d** for Pd@P ↑ -In₂Se₃). **e** Simplified schematic diagrams of CO₂ bonding to transition metals.

Supplementary Fig. 18) by forming the bidentate C-TM-O species. The only exception is Zr@P ↓ -In₂Se₃, where the linear O=C=O structure is well preserved (only bent by 0.8°) because of the monodentate TM-O configuration. Moreover, the binding energies of the catalysts with the activated CO₂ molecule (−3.39 to −0.76 eV; see Table 1) are comparable to the reported catalysts[55,61,62]. The strong hybridization of O $p$ and TM $d$ orbitals and the significant charge transfer between TM@In₂Se₃ and CO₂ ensure the inert molecule chemically captured (see Fig. 2b, d, and Supplementary Fig. 19). Note that the degree of CO₂ activation on two opposite surfaces are obviously different as seen from the angles of ∠OCO (see Table 1), which are generally smaller when the molecule is adsorbed on the surface with polarization downwards. Meanwhile, the activation degree of CO₂ on TM@In₂Se₃ (TM = Nb, Re) is significantly higher than that on TM@In₂Se₃ (TM = Ni, Pd, Rh), as indicated by the larger binding energies (−3.39~−2.02 eV vs −1.39~−0.76 eV), more electron transfer (0.61~0.92e vs 0.26~−0.45e), and smaller angles of ∠OCO (131.1°~136.8° vs 146.4°~151.2°).

The distinct behaviours on different ferroelectric surfaces are attributed to the mechanism of CO₂ catalytic activation, namely coexistence of charge depletion and accumulation between the TM atoms and CO₂ molecules. The empty TM $d$ orbital could accept electrons from the highest occupied molecular orbitals ($1\pi_g$ orbital) of the CO₂ molecule, while the occupied TM $d$ orbital could back-donate electrons to the lowest unoccupied molecular orbitals ($2\pi_u$ orbital) of the CO₂ molecule (Fig. 2e). As indicated in Fig. 2b and d, the $d$ orbital occupations of TM atoms on the ferroelectric In₂Se₃ layer can be effectively modulated by the switchable polarization. The synergy of electron acceptance, back-donation, and polarization dependent $d$ orbital occupation ensures that CO₂ can be efficiently activated, while the activation degree is ferroelectric controllable in TM@In₂Se₃ catalysts[61]. For example, while isolated Pd atom has fully occupied $d$ orbital ($4d^{10}$), upon adsorption on the In₂Se₃ substrate, the Pd atom loses partial $d$ electrons to form the Pd-Se bond, and the resulting empty Pd $d$ orbitals provide channels for electron acceptance and back donations to activate CO₂ molecules. Consequently, compared with other TM@In₂Se₃ catalysts, where the TM atoms have intrinsic partially occupied $d$ orbitals, Pd@In₂Se₃ catalyst has a lower degree of CO₂ activation as indicated by the relatively larger ∠OCO and smaller charge transfer (see Table 1). Furthermore, for TM@In₂Se₃ under polarization switching, the density of states (DOS) variations are more evident than the CO₂

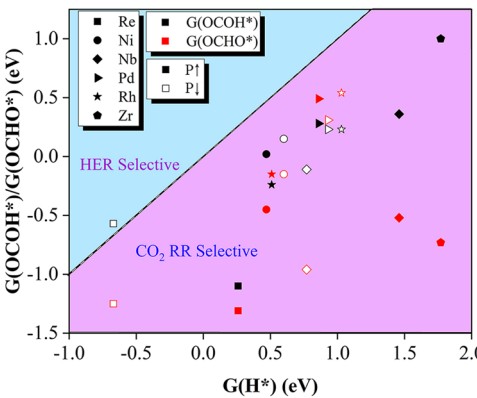

**Fig. 3 Selectivity for CO₂RR vs HER.** Gibbs free energy changes (ΔG) of initial protonation of CO₂RR vs. HER on TM@In₂Se₃ (TM = Ni, Pd, Rh, Zr, Nb and Re). Data points in the purple region denote higher selectivity toward CO₂RR, while those in the blue region toward HER. Black and red symbols stand for ΔG(OCOH*) vs ΔG(H*) and ΔG(OCHO*) vs ΔG(H*), respectively.

binding energy and electron transfer. This is due to the fact that the DOS variation is directly related with the orbital shift under the polarization flip, such as the $d$ band center, as seen in Fig. 1c, while the charge transfer and the CO₂ binding energy are mainly related to the itinerant electrons induced by the polarization.

**Selectivity for CO₂RR vs HER.** As an important competing side reaction, hydrogen evolution reaction (HER) may significantly restrain the Faradaic efficiency of CO₂RR by consuming proton–electron pairs from the electrolyte solution[63,64]. To check if CO₂RR is more favourable, we first calculated Gibbs free energy changes (ΔG) at the first hydrogenation step of CO₂RR (* +CO₂ + H⁺ +e⁻ → OCOH* or OCHO*) and HER (* +H⁺ +e⁻ → H*). According to the Brønsted–Evans–Polanyi relation[65,66], reactions with lower ΔG have smaller reaction barriers and consequently are kinetically more favoured.

As shown in Supplementary Fig. 20, for the first hydrogenation step of CO₂RR, Rh@In₂Se₃, Pd@In₂Se₃, and Re@P ↑ -In₂Se₃ prefer to form carboxyl (OCOH*), while Ni@In₂Se₃, Nb@In₂Se₃, Zr@P ↑ -In₂Se₃, and Re@P ↓ -In₂Se₃ tend to produce formate (*OCHO). Since CO₂ molecule cannot be activated on Zr@P ↓ -In₂Se₃, it is not considered here. The Gibbs free energy changes

(Fig. 3) indicate that all the screened TM@In$_2$Se$_3$ catalysts have a higher selectivity toward CO$_2$RR. These results demonstrate the feasibility of using TM@In$_2$Se$_3$ catalysts as cathodes for CO$_2$RR with a high Faradaic efficiency.

**Tunable CO$_2$RR by polarization switching**. As a result of polarization dependent $d$ band center (see Fig. 1c), CO$_2$ molecules are activated to different degrees depending on the ferroelectric surfaces. It is thus expected that the reaction barrier, reaction path, and even the intermediate/final product of CO$_2$ reduction will be polarization dependent and controllable via ferroelectric switching.

Considering the complexity of CO$_2$ reaction, we comprehensively searched the minimum energy reaction paths of CO$_2$RR on each TM@In$_2$Se$_3$ (TM = Ni, Pd, Rh, Zr, Nb and Re) catalyst (Supplementary Fig. 20). Table 2 summarizes the effects of the

polarization switching on the CO$_2$RR, from the aspects of potential determining steps, limiting potential ($U_l$), and final product.

Depending on the effects of the reversible polarization, the ferroelectric catalysts for CO$_2$RR can be classified into three categories: (i) TM@In$_2$Se$_3$ (TM = Ni, Pd, Rh), for which the polarization switching changes the reaction barriers, but not the reaction paths and the intermediate/final products; (ii) Zr@In$_2$Se$_3$, for which the ferroelectric switching can reactivate the stuck CO$_2$ reduction; (iii) TM@In$_2$Se$_3$ (TM = Nb, Re), for which all the reaction paths, reaction barriers, and final products are polarization dependent. This classification is well consistent with $d$ band center analysis (Fig. 1c), as well as the binding energy, charge transfer, and activation degree of CO$_2$ as listed in Table 1, which indicates clear underlying relationship among these material properties and catalytic performance.

**Tunable limiting potential of CO$_2$RR (on TM@In$_2$Se$_3$, TM = Ni, Pd, Rh)**. Since the activation degree of CO$_2$ on the ferroelectric surface depends on polarization direction due to the adjusted empty and occupied $d$-orbitals (shifted $d$ band center), the reaction barrier of the CO$_2$ hydrogenation process is also affected. Taking the Rh@In$_2$Se$_3$ catalyst as an example (see Fig. 4a), though the configurations of activated CO$_2$ (with C atom connecting with Rh atom) under different polarization directions are the same, the energy barriers are obviously different: the reaction of CO$_2^*\rightarrow$OCOH$^*$ is downhill on Rh@P$\uparrow$-In$_2$Se$_3$, but uphill on Rh@P$\downarrow$-In$_2$Se$_3$. In the subsequent hydrogenation steps, the polarization reversal does not change the most favourable path of CO$_2$RR, both following the paths of CO$_2^*\rightarrow$OCOH$^*\rightarrow$CO$^*\rightarrow$CHO$^*\rightarrow$CH$_2$O$^*\rightarrow$CH$_2$OH$^*\rightarrow$CH$_3$OH$^*\rightarrow$OH$^*\rightarrow$H$_2$O$^*$, with the final product of CH$_4$. However, the reaction barrier for each step, the potential-determining step

**Table 2 Potential determining steps (PDS), limiting potentials ($U_l$, V vs RHE), and final product of CO$_2$RR with TM@In$_2$Se$_3$ catalysts.**

| Catalyst | PDS | $U_l$ | Product |
|---|---|---|---|
| Ni@P$\downarrow$-In$_2$Se$_3$ | HOCHO$^*\rightarrow$CHO$^*$ | −0.36 | CH$_4$ |
| Ni@P$\uparrow$-In$_2$Se$_3$ | HOCHO$^*\rightarrow$CHO$^*$ | −0.47 | CH$_4$ |
| Pd@P$\downarrow$-In$_2$Se$_3$ | CO$^*\rightarrow$CHO$^*$ | −0.77 | CH$_4$ |
| Pd@P$\uparrow$-In$_2$Se$_3$ | CO$^*\rightarrow$CHO$^*$ | −0.87 | CH$_4$ |
| Rh@P$\downarrow$-In$_2$Se$_3$ | CO$^*\rightarrow$CHO$^*$ | −0.43 | CH$_4$ |
| Rh@P$\uparrow$-In$_2$Se$_3$ | CHO$^*\rightarrow$CH$_2$O$^*$ | −0.26 | CH$_4$ |
| Zr@P$\downarrow$-In$_2$Se$_3$ | HOCHO$^*\rightarrow$CHO$^*$ | −2.33 | CH$_4$ |
| Re@P$\downarrow$-In$_2$Se$_3$ | OH$^*\rightarrow$H$_2$O$^*$ | −0.65 | CH$_3$OH |
| Re@P$\uparrow$-In$_2$Se$_3$ | COH$^*\rightarrow$C$^*$ | −0.85 | CH$_4$ |
| Nb@P$\downarrow$-In$_2$Se$_3$ | OH$^*\rightarrow$H$_2$O$^*$ | −0.64 | CH$_3$OH |
| Nb@P$\uparrow$-In$_2$Se$_3$ | OCHO$^*\rightarrow$HOCHO$^*$ | −0.58 | HOCHO |

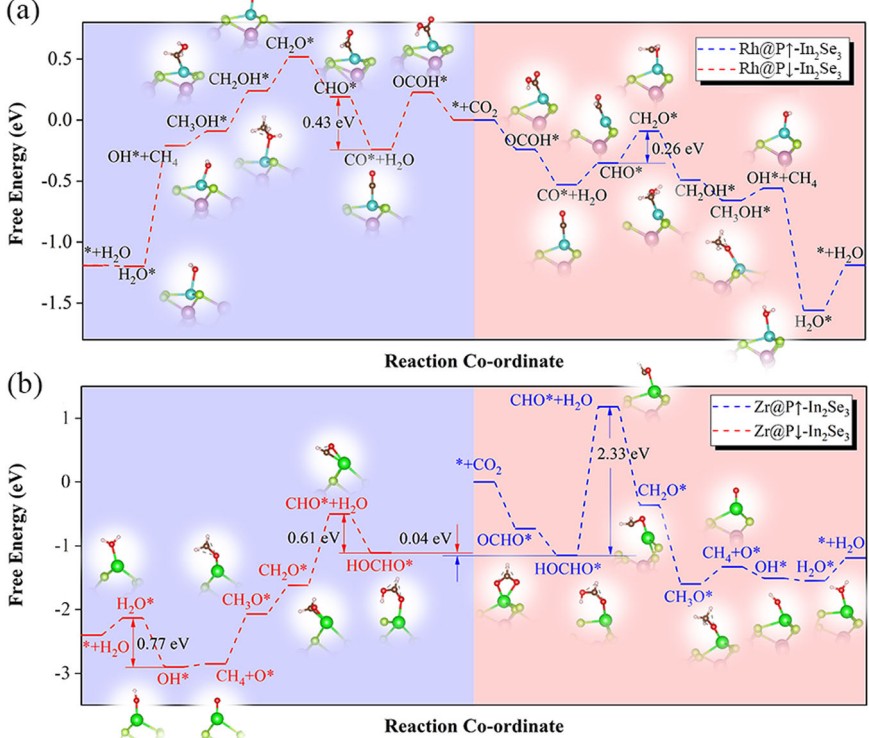

**Fig. 4 CO$_2$RR paths on Rh@In$_2$Se$_3$ and Zr@In$_2$Se$_3$.** The free-energy profile for the CO$_2$ electrochemical reduction reactions along the minimum energy path at 0 V (vs. RHE) on **a** Rh@In$_2$Se$_3$ and **b** Zr@In$_2$Se$_3$. The insets show the optimized configurations of the intermediates. The pink (light blue) shaded area indicates the catalytic reaction when the polarization is pointing upwards (downwards). This color scheme is also used in Fig. 5.

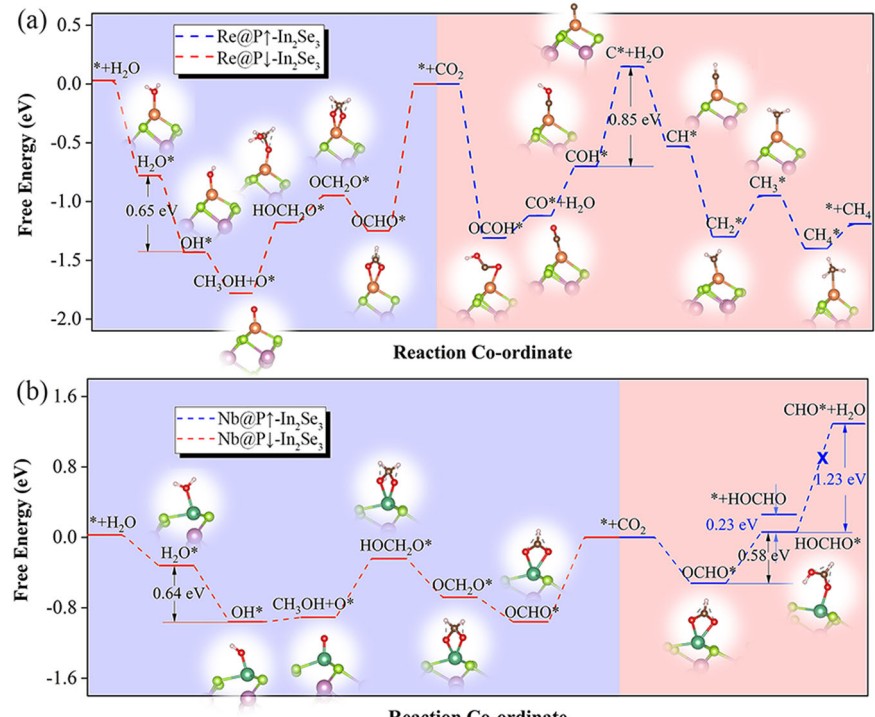

**Fig. 5 CO₂RR paths on Re@In₂Se₃ and Nb@In₂Se₃.** Free energy profile for CO₂ electrochemical reduction reactions along the minimum energy path at 0 V (*vs.* RHE) on **a** Re@In₂Se₃ and **b** Nb@In₂Se₃. The insets show the optimized configurations of the intermediates.

(PDS), and the limiting potential are all different and polarization dependent. On Rh@P↓-In₂Se₃, the PDS of CO₂ reduction is CO* →CHO* with a limiting potential of −0.43 V *vs.* RHE; in contrast, the corresponding PDS on Rh@P↑-In₂Se₃ is the protonation of CHO* to CH₂O* with the estimated $U_l$ of −0.26 V *vs.* RHE. Obviously, the reaction barrier of CO₂ reduction is polarization tunable via ferroelectric switching. Notably, the limiting potential (−0.26 eV *vs.* RHE) is better than most reported CO₂RR catalysts (listed in Supplementary Table 6). Therefore, Rh@In₂Se₃ is potentially an efficient electrochemical catalyst for CO₂RR with controllable performance. Similar phenomena of polarization dependent catalysis also exist in Ni@In₂Se₃ and Pd@In₂Se₃ (Table 2 and Supplementary Fig. 21). The relatively weaker but still obvious ferroelectric dependent CO₂ reduction relates with the moderately polarization dependent *d* band center (−1.98∼ −1.16 eV at P↓; −4.13∼−3.16 eV at P↑, Fig. 1c) and modest activation degree of CO₂, as indicated by the moderate CO₂ binding energies ($E_{b-CO2}$, 0.79-1.39 eV) and mild charge gained by the adsorbed CO₂ molecule ($Q_{CO2}$, 0.26-0.45e) (Table 1).

**Reactivating stuck catalytic reaction of CO₂RR (on Zr@In₂Se₃).** Zr@P↓-In₂Se₃ cannot activate CO₂ molecules, it is thus unsuitable for CO₂ reduction. In contrast, CO₂ can be activated on Zr@P↑-In₂Se₃ with two oxygen atoms bridging over Zr atom (Supplementary Fig. 18i, j). With more protons, it follows the hydrogenation steps of CO₂RR as * + CO₂ + H⁺ +e⁻ → OCHO* and OCHO* +H⁺ +e⁻ → HOCHO*; both of these steps are downhill, indicating that the reduction can react easily. However, further hydrogenation step is prevented by the strong binding strength of the intermediate HOCHO* (ΔG = 2.33 eV, Fig. 4b) due to significant electron transfer and over-activations (1.13 e between CO₂ and catalyst, 126.2° of ∠OCO as shown in Table 1). Based on the Sabatier principles, such strong binding energies make further reduction only possible at very negative potentials. Further hygenation is thus prehibitted.

Since polarization can modulate interaction strength as indicated above, it is expected that the binding strength of the intermediate HOCHO* can be weakened when the polarization direction of In₂Se₃ is reversed, so that the catalytic reduction can proceed. Indeed, we found that when polarization direction is reversed from up to down at this step, the barrier decreases (ΔG = 0.61 eV for the reduction of HOCHO* to CHO* on Zr@P↓-In₂Se₃). More importantly, HOCHO* on the Zr@P↓-In₂Se₃ is merely 40 meV higher than that on Zr@P↑-In₂Se₃, which indicates that the polarization switching is relatively easy to achieve. The conclusion holds true under dilute Zr concentrations, while the local ferroelectric switching around HOCHO* is prohibited due to the higher energy barrier (see Supplementary Fig. 22). Additionally, according to the subsequent hydrogenation steps, the $U_l$ for Zr@P↓-In₂Se₃ is −0.77 V *vs.* RHE, which is significantly lower than that for Zr@P↑-In₂Se₃ (−2.33 V *vs.* RHE). Moreover, the H₂O* at Zr@P↓-In₂Se₃ could be removed spontaneously, in contrast to the 0.36 eV energy barrier for the dehydration process on Zr@P↑-In₂Se₃.

Therefore, though CO₂ molecule cannot be activated on Zr@P↓-In₂Se₃ for reduction, the polarization reversal can reactivate the catalytic reduction on Zr@P↓-In₂Se₃ with a reasonable limiting potential. During the hydrogenation process, appropriate polarization switching could either reactivate the CO₂ reduction or accelerate the reaction rate by reducing the reaction barriers.

**Tuning CO₂RR path and final product (on Nb@In₂Se₃ and Re@In₂Se₃).** In contrast to the aforementioned TM@In₂Se₃ (TM = Ni, Pd, Rh), the activations of CO₂ molecule are much deeper on Nb@In₂Se₃ and Re@In₂Se₃, as evidenced by the smaller bond angle ∠OCO (Table 1), much larger $E_{b-CO2}$ values (2.02-2.54 eV), and more pronounced charge transfer between CO₂ and the catalyst ($Q_{CO2}$, 0.61-0.92e). The huge differences of *d* band center (−2.52∼−2.52 eV at P↓; −7.03∼−6.72 eV at P↑, see Fig. 1c) render the catalytic reaction more sensitive to the

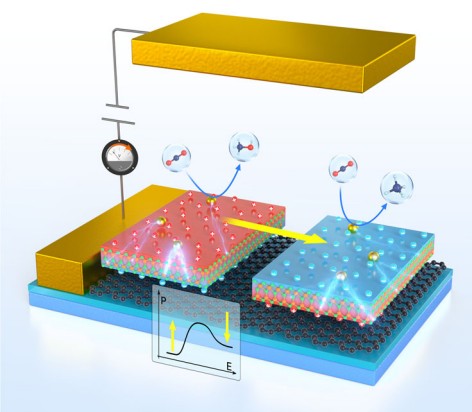

**Fig. 6 Schematic diagram of ferroelectric controllable CO₂RR.** Here, metal anchored α-In₂Se₃ monolayer is placed between the electrodes to achieve ferroelectric switching and controllable catalysis, tuned by the reversal of the bias direction.

polarization switching, leading to different reaction paths and even different final products according to the Sebastian principles.

Indeed, the polarization switching in the catalysts of Nb@In₂Se₃ and Re@In₂Se₃ could partially or even totally alter the CO₂RR path, thus leading to the different final products. As illustrated in Fig. 5a, for CO₂RR on Re@P↓-In₂Se₃ catalyst, the minimum energy path is $CO_2^* \rightarrow OCHO^* \rightarrow OCH_2O^* \rightarrow HOCH_2O^* \rightarrow O^* \rightarrow OH^* \rightarrow H_2O^*$, the final product is methanol, and the PDS is $OH^* \rightarrow H_2O^*$ with $U_l$ of 0.65 eV; In sharp contrast, the reaction paths on Re@P↑-In₂Se₃ is $CO_2^* \rightarrow OCOH^* \rightarrow CO^* \rightarrow COH^* \rightarrow C^* \rightarrow CH^* \rightarrow CH_2^* \rightarrow CH_3^* \rightarrow CH_4^*$, the final product is methane, and the PDS is $COH^* \rightarrow C^*$ with $U_l$ of 0.85 eV. Similar CO₂RR phenomena were also observed on Nb@In₂Se₃ catalyst. The most favourable path on Nb@P↓-In₂Se₃ surface is $CO_2^* \rightarrow OCHO^* \rightarrow OCH_2O^* \rightarrow HOCH_2O^* \rightarrow O^* \rightarrow OH^* \rightarrow H_2O^*$, producing methanol with $U_l$ of 0.64 eV; in comparison, the $CO_2^* \rightarrow OCHO^* \rightarrow HOCHO^*$ path on Nb@P↑-In₂Se₃ produces methanoic acid with $U_l$ of 0.58 eV (Fig. 5b).

Note that selective generation of desired chemical fuel products has been pursued in the development of electrochemical CO₂RR catalyst over decades[5], our strategies via feasible ferroelectric switching can achieve the goal without complex and high-cost steps that use different catalysts or electrolytes to produce desired fuels[67–70]. As shown in Fig. 6, our proposed ferroelectric SACs base on experimentally available In₂Se₃ and unique polarization dependent catalytic performance thus open an avenue for controllable CO₂ reduction.

## Discussion

In conclusion, by comprehensive DFT computations, we have screened out six TM@In₂Se₃ (TM = Ni, Nb, Pd, Re, Rh and Zr) as effective ferroelectric catalysts for electrochemical CO₂RR. We found that the polarization reversal of ferroelectric In₂Se₃ monolayer can adjust the empty/occupied $d$ orbitals of adsorbed TM atom and the energy of the $d$ electrons, thereby tuning the catalytic performance for CO₂RR, including the degree of CO₂ activation, limiting potential, reaction path, and final product, even reactivating stuck catalytic reduction. Especially, the Rh@P↑-In₂Se₃ catalyst is indentified as a highly efficient electrochemical catalyst for CO₂RR due to the fairly low limiting potential (<0.5 V)[71]. Moreover, the Re@In₂Se₃ and Nb@In₂Se₃ catalysts can realize selective generation of different desired products with the same catalyst via polarization switching, which opens a path toward ferroelectric controllable catalysis. These SACs based on

2D ferroelectric materials hold great promise for improving catalytic activity and selectivity for electrochemical CO₂RR.

## Methods

**Computational details.** Our density functional theory (DFT) computations are performed with the Vienna ab initio simulation package (VASP) code[72,73]. The spin-polarized generalized gradient approximation (GGA) in the form of Perdew − Burke − Ernzerhof (PBE) treats the exchange − correlation interactions, while the frozen-core projector augmented wave (PAW) approximation describes the interaction between the ion and electron[74–76]. The van der Waals (vdW) interactions are described with the DFT-D3 method in Grimme's scheme[77]. The models with one TM atom anchored 2 × 2 In₂Se₃ are built as the potential catalysts for the simulations, since it corresponds to the coverage of 25% with the optimal adsorption energies, as shown in Supplementary Fig. 23. Four TM atoms uniformly distributed on the hexagonal centers of 4 × 4 In₂Se₃ supercell act as the catalytic active sites. More than 20 Å vacuum space perpendicular to the surface is added to avoid interactions between periodic slabs. The dipole correction is taken into account for all the asymmetric structures[78]. The 2D Brillouin sampling is presented with a 4 × 4 × 1 gamma-centered Monkhorst-pack $k$-mesh. The cutoff energy for plane-wave basis sets is 500 eV. The convergence thresholds for force and total energy are $10^{-2}$ eV/Å and $10^{-5}$ eV, respectively. Moreover, the Gibbs free energy calculation is operated with computational hydrogen electrode (CHE) model[79], and the solvent effect is considered with the implicit solvent model implemented in VASPsol[27,61]. The site-specific charge differences are obtained using Bader analysis. Besides the standard PBE calculations and the models of TM@2 × 2 In₂Se₃, we also reexamined the chemical reactions by using larger supercells (TM@4 × 4 or 6 × 6 In₂Se₃, Supplementary Fig. 24) or different exchange functions (PBE + U or RPBE; Supplementary Figs. 25 and 26). All these approaches lead to the same conclusion about the ferroelectric controlled CO₂ reduction, with the key numerical results only slightly different, indicating the robustness of our conclusion. More details of the simulations can be found in the Supplementary Information.

## Data availability

Source data are provided with this paper. The data used in this study are presented in the text and Supplementary Information. Additional data and information are available from the corresponding author upon reasonable request. Source data are provided with this paper.

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

## Acknowledgements

We are grateful to Prof. Dongwei Ma from Henan University for discussing the mechanism of electrocatalysis. We acknowledge the grants of high-performance computer time from the computing facilities at the Queensland University of Technology, the Pawsey Supercomputing Center, and the National Computational Infrastructure (NCI) facility at the Australian National University allocated through both the National Computational Merit Allocation Scheme supported by the Australian Government and the Australian Research Council Grant LE190100021 (Sustaining and strengthening merit-based access at NCI, 2019 − 2021). L.J. acknowledge the support through National Natural Science foundation of China (Grants No. 11804006), Henan Key Program of Technology Research and Development (No. 182102310907), Henan College Key Research Project (No. 19A430006), and the China scholarship council for its financial support (No. 201908410036); Y.G. acknowledges the support by ARC Discovery Project DP200102546; Z. C acknowledges the support by the National Science Foundation-Centers of Research Excellence in Science and Technology (NSF-CREST Center) for Innovation, Research and Education in Environmental Nanotechnology (CIRE2N) (Grant No. HRD-1736093).

## Author contributions

L.K. proposed, designed the project and organized the paper. L.J. and X.T. conducted the simulations, data analyses and wrote the manuscript, L.J. and X.T. made equal contributions to the work. X.M., Y.G., S.S., A.D., Z.C. and C.C. contributed to the discussion of results and manuscript revision.

## Competing interests

The authors declare no competing interests.
