## [Peer Review File · Nature Communications]

REVIEWER COMMENTS

Reviewer #1 (Remarks to the Author):

The authors present a very nice and complete study of the effect of using single-atom catalysts based on several transition metals on a ferroelectric In₂Se₃ support. They show that switching the ferroelectric polarization could, depending on the transition metal, be an effective way to reduce the limiting potential for CO₂ reduction, reactivate CO₂ reduction or affect the selectivity of the catalyst. First of all, I very much enjoyed reading this very interesting and well-structured manuscript. The work is performed in a very solid fashion, the reported catalysts are highly active and selectivity as one of the main issues in CO₂ reduction is addressed. These points along with the fact that ferroelectric catalysis is a currently emerging hot topic, the manuscript is appropriate for Nature Communications and I recommend that the manuscript be published after the authors address the following comments:

Could the authors hypothesize how a practical device based on their catalyst would be constructed? How is the ferroelectricity switched? Does the increased efficiency outweigh the energetic cost for FE switching?

The authors always write "the ferroelectric switch". The term "ferroelectric switching" is more commonly used and I would suggest the authors to change all occurrences (starting with the title).

Figure 1 would be clearer if cell boundaries would be shown (at least for the in-plane directions) such that the periodicity and viewing direction is more obvious. Also could the authors clarify if the charge density differences in Figure 1f are computed at the same geometry or if the density difference could be affected by atomic-relaxation artefacts? This is unfortunately not clear from the corresponding methods section in the SI.

I find the MD data shown in figure S6 quite weak as a support for the absence of structural changes. In that figure we merely see that the energy fluctuates ~ 0.05 eV but what these fluctuations correspond to remains completely unclear. The authors should show multiple configuration snapshots (at the very least two, one at the beginning and one at the end).

In Figure 2 is the energy axis $E-E_{\text{Fermi}}$? If yes, this could be clarified in the figure. Also should it not be "sigma back donation"? It looks like sigma overlap to me.

When performing the analysis of the CO₂RR pathways, how were configurations screened? I.e. did the authors test multiple adsorption sites and adsorption modes? If yes, how many and how were they generated/selected?

I find Figure 4 a little confusing, in particular because different representations are used in a) and b). To be clearer, it would be nice if panel b adopted the same "to-the-left" and "to-the-right" scheme as panel a). This scheme could be further clarified by more clearly marking the starting point in the center.

The authors use PBE, would PBE+U for the TM d states change anything in the results? Also I believe there is a typo "gamma-pack" in the methods section that should be "gamma-centered Monkhorst-Pack".

While the article is written quite well, the text could benefit from thorough proofreading, ideally by a native speaker, as some conjugation mistakes, missing articles and other grammar issues exist. More particularly, the authors should try to use clearer formulations in places such as "does not have enough electrons" (page 2, top of right column), "high ratio of low-coordinates configurations" (page 2, middle of right column), "ensure the inert molecule chemically captured" (page 4, bottom of left

column), "substract" should be substrate (page 5, top of left column), "depending on the ferroelectric surfaces" maybe use "ferroelectric state of the surface" (page 5, middle of right column), "the activations ... are much deeper" (page 7, top of left column).

Reviewer #2 (Remarks to the Author):

This work reports a theoretical investigation of single-atom catalysts based on ferroelectric alpha-In₂Se₃ for CO₂ reduction. As the authors expected, alpha-In₂Se₃ may be applicable to CO₂ reduction. However, I still have some concerns about the stability of In₂Se₃ itself, which has to be ensured, particularly if the authors consider actual applications (see detailed comments below). Thus, I do not think that this work qualified enough to be published on Nature Communications.

Major concerns:

1. In this work, the authors mainly focused on how stable single metal atoms are on alpha-In₂Se₃. However, the more critical concern is the stability of the substrate itself. Compounds consisting of In and Se are possible to have various compositions and structural polymorphs that can appear at typical atmospheric conditions like room temperature and atmospheric pressure. Thus, In-Se compounds have been studied for application to phase change memory. To use materials as electrochemical catalysts for CO₂R, the stability of the materials must be ensured especially under harsh environments like alkaline or acidic aqueous solutions. However, to the best of my knowledge, alpha-In₂Se₃ has not been proven to be stable and I am skeptical to its stability in light of the presence of various competing form of In-Se compounds. Furthermore, single atom catalysts need significant doping of alpha-In₂Se₃, but such a doping would deteriorate the phase stability of the material as well.
2. As potential single-metal-atom sites, several atomic sites on the surface are accounted for (i.e., the dopants are assumed to act as interstitial defects), but introduced dopants may prefer to occupy other sites (e.g., substitution for In sites). To determine the most plausible atomic configurations of the single-metal-atoms, the thermodynamic defect formation energy should be checked, taking the chemical potentials of defect components into account.
3. When calculating binding energies of single metal atoms, which energy reference is considered for isolated metal atoms? For instance, Ref. 53 considered the energy per atom of the stable bulk metal phase when calculating the binding energies.
4. 5-ps MD simulations at 300 K (I assume that the authors used NVT ensemble) may provide a meaningful insight into the stability of single metal atoms, but it does not give useful information for the overall stability of the catalyst including the support.
5. The authors should mention how large supercells (i.e., single-metal-atom concentrations) are used. My feeling is that the dopant concentration considered in the present work is too large to occur in actual materials because the DOS of alpha-In₂Se₃, which is a semiconductor (PBE gap is around 0.4 eV?), is highly metallic when it is doped. Because the catalytic activity is likely to depend on the single-metal-atom concentration, the authors should first discuss the catalytic activity when the doping concentration is in dilute limit.
6. In Figure S8, the energy difference between the down and up polarizations is only ~0.15 eV/UC. The value seems to me inconsistent with the binding energy difference of Pd depending on the polarization direction (-1.69 eV/atom vs. -1.14 eV/atom).
7. I found that even if the DOS of the transition metal anchored on alpha-In₂Se₃ significantly varies depending on the polarization direction, the charge lost and CO₂ binding energy remains almost the same. Taking Pd@In₂Se₃ as an example, QTM = 0.14 and 0.12 and Eb-CO₂ = -0.84 and -0.79 eV for the down and up polarizations, respectively. This was not clearly explained.
8. Regarding Zr@In₂Se₃ catalysts, they mentioned that the ferroelectric switching can readily occur when HOCHO* exists. However, the calculation within periodic boundary condition enforces the ferroelectric switching that occurs across the entire lattice. In actual catalysts, single metal atoms are likely to be fairly apart from each other and such a ferroelectric switching is difficult to occur. Local ferroelectric switching around HOCHO* may occur, but in general, the local ferroelectric switching is energetically prohibitive.

9. In this work, PBE functional is used, but the free-energy diagram depends on a type of exchange-correlation functionals. Thus, the direct comparison of their results with other calculations is unfair. Moreover, RPBE or BEEF functionals are known to be more relevant for studying molecular adsorptions. They need to check if their conclusion changes when using such functionals.

Minor comments:

1. How did the authors calculate site-specific charge differences like QTM and QCO₂? Did they use Bader analysis?
2. In Table S3, the entropies of liquid species are presented. How these values are obtained?
3. Besides the out-of-plane polarization, the direction of the in-plane polarization should be marked.

Reviewer #3 (Remarks to the Author):

This paper presents a thorough first-principles study of single atom catalysts (SACs) for CO₂ electroreduction supported by ferroelectric In₂Se₃ semiconductor. This is certainly an important topic and the authors identify the optimal metal elements for In₂Se₃-supported SACs and also show how the change in ferroelectric polarization can be used to strongly modify the catalytic properties of these systems. However, to demonstrate that their results are of a broad interest, the authors must show or at least provide arguments for the superiority of the proposed SAC-In₂Se₃ catalysts to the state of the art catalysts reported in recent work for one or more of the following: lower overpotential, greater selectivity, lower cost, ease of fabrication, etc. In other words, why should the community invest effort in trying to experimentally realize this system instead of using the more standard SAC substrates such as carbon?

Technically, I think the authors must address the following issues.

1) The coupling of ferroelectricity and catalysis is certainly interesting. However, a monolayer In₂Se₃ with adsorbed metal atoms may have significantly different electronic structure, for example becoming metallic which would preclude the switching of the polarization by an applied electric field. By contrast, a thin layer of metal on top of a thick FE oxide material studied in previous work will still be an insulator in the bulk and therefore will show switching behavior. The authors must address this point and demonstrate that the adsorption of SAC does not destroy the ferroelectricity (that is the switchability of polarization with electric field) of the In₂Se monolayer system.

2) To demonstrate the stability of SAC with respect to agglomeration, the authors perform ab initio MD simulations at 300 K for 5 ps. This is not sufficient for exploring the potential energy surface and demonstrating that SAC will not diffuse because diffusion may take place on a time scale much longer than 5 ps. Therefore, a different approach must be used to demonstrate stability versus agglomeration (e.g. agglomeration reaction pathway calculations, or perhaps longer and/or higher temperature simulations).

3) For electrocatalysts, good conductivity of the electrode/catalyst is typically important. Since In₂Se₂ is a semiconductor, will this tend to decrease the electroreduction performance?

Considering the above, while the authors do a good job proposing a new class of SAC CO₂ reduction catalysts, it is not clear from the present manuscript that this class is indeed promising and enables achievement of performance that is unavailable or even matches the performance obtained using other classes of SAC systems or that it provides a new mechanism for catalysis. If the authors make a convincing case for the promise of these catalysts the paper may be suitable for publication in Nature Communications.

REVIEWER COMMENTS

Reviewer #1 (Remarks to the Author):

The authors have addressed all my previous concerns in a satisfactory fashion and have improved the manuscript considerably.

Reviewer #2 (Remarks to the Author):

Authors have addressed many concerns raised by the reviewers, but more clarifications are needed to recommend for publication.

1. I did not see how many dopants can be incorporated in interstitial sites. It is better to explicitly calculate the formation energy of interstitial dopants in order to show SAC based on In₂Se₃ can actually have a large number of active sites.
2. Taking the energy of an isolated atom as a reference in calculations of binding energy does not reflect experiments wherein molecular precursors are used to dope systems. An isolated metal atom is definitely less stable than the metal in a molecule. For the binding energy to have proper physical meaning, the authors should set an adequate energy reference. Otherwise, we cannot judge the binding strength based on the binding energy. In addition, using an isolated atom as a reference is likely to lead to fairly large binding energies of TMs because of instability of an isolated atom.

Reviewer #3 (Remarks to the Author):

The authors have answered most of the criticisms from the previous review round. However, two issues are still unresolved. The more important is the possible agglomeration of the Pd atoms into nanoparticles. The authors have now performed 15 ps AIMD simulations to demonstrate that agglomeration is not favorable. These simulations are only marginally more useful than the 5 ps AIMD simulations performed previously because the agglomeration may occur on the ns or even microsecond timescale. So extending the simulation timescale from 5 ps to 15 ps is insufficient. I understand that AIMD simulations on ns timescale are impossible. Therefore, a different method must be used to demonstrate the lack of agglomeration. The authors take a step towards such a demonstration in their calculations of the dimer energy versus the single-atom Pd adsorption. However, even though the dimer is higher in energy than single atom, this does not yet prove that agglomeration is thermodynamically unfavorable. For example, while a dimer may be unfavorable, a particle where most of Pt atoms are bound to other Pt atoms rather than In₂Se₃ surface (i.e. a 3D Pd particle) may be favorable so that the dimer state would represent a possible barrier state but does not fully describe the thermodynamic potential surface of this system.

For a complete description of this problem, a kinetic Monte Carlo simulations would be a good method. I am not sure if this is required in this case, but something more sophisticated than dimer energy calculation and 15 ps AIMD is certainly necessary.

Second, with regard to the issue of stability of alpha-In₂Se₃ under electrochemical environment and doping, it is not enough to consider the small perturbation of the structure under doping because the structure may be trapped in the alpha-In₂Se₃ local minimum, whereas in experiment other phases may be the global energy minimum in the presence of doping and electrochemical potential and possible effects of H⁺ and counterions. Thus, to demonstrate that alpha-In₂Se₃ is the global minimum energy phase even in the presence of dopants and in electrochemical environment, the other possible

In/Se phases with dopants and in the presence of H, OH and possibly counterions should be compared to α -In₂Se₃ with dopants and in the presence of H, OH and possibly counterions. I think this point is less crucial than the question of agglomeration because even if another phase is the global minimum in the presence of dopants, the α -In₂Se₃ may still be kinetically trapped and stable. Nevertheless, this still should be addressed

REVIEWERS' COMMENTS

Reviewer #3 (Remarks to the Author):

The authors have successfully addressed my concerns regarding their evaluation of the stability of the SAC and the alpha-In₂Se₃ phase. The paper is now suitable for publication in Nature Communications.

Changes made in the revised manuscript (the revised texts have been highlighted in red):

1. The advantages of ferroelectric SACs have been highlighted in the Abstract, Introduction, and Conclusion sections.
2. The issues related to the stability of the ferroelectric α -In₂Se₃ layers have been discussed at the beginning of “Results and Discussion” section.
3. Discussions on possible metal substitution as catalysts have been added in page 2.
4. Figure 1 has been updated, with the cell boundaries added and supercell size indicated.
5. The stability of TM@In₂Se₃ from AIMD simulations and clustering calculations have been further discussed in the right column of page 3.
6. It has been pointed out (see the right column of page 4) that the metal adsorption does not affect the ferroelectricity of In₂Se₃ monolayer.
7. The difference between the DOS variation and electron transfer & binding energy has been explained in the right column of page 5.
8. The prohibition of local ferroelectric switching has been stated in the left column of page 7.
9. Figure 4 has been updated.
10. Figure 6 has been added to guide possible experimental verification.
11. Discussions have been added in the left column of page 8 to corroborate the robustness of our findings (supercell size and methods).
12. The author list has been updated.
13. The description of the SI has been updated.
14. The presentation, including the use of English language, has been improved.

Changes made in the revised Supporting Information (revisions are highlighted in red):

1. Discussions on the structural stability of α -In₂Se₃ layers have been added in pages 1-3, with two new figures inserted.
2. Discussions on the stability of α -In₂Se₃ layers under harsh environments have been added in page 4.
3. Discussions and two associated figures regarding the possibilities of metal substituted In₂Se₃ have been added in pages 6-7.
4. Updated AIMD results have been added in page 10.
5. Calculations of clustering energies and discussions have been added in page 11.
6. Discussions on local metallization are presented in pages 13-14.
7. Discussions on local vs whole ferroelectric switching have been added in page 18.
8. Discussions on the dependence of catalytic activity on metal concentrations and related Figure S19 have been added in page 34.
9. Discussions on PBE vs PBE+U (RPBE) calculations have been added in pages 35-37.

Responses to the Reviewers' Comments

Reviewer #1 (Remarks to the Author):

The authors present a very nice and complete study of the effect of using single-atom catalysts based on several transition metals on a ferroelectric In₂Se₃ support. They show that switching the ferroelectric polarization could, depending on the transition metal, be an effective way to reduce the limiting potential for CO₂ reduction, reactivate CO₂ reduction or affect the selectivity of the catalyst. First of all, I very much enjoyed reading this very interesting and well-structured manuscript. The work is performed in a very solid fashion, the reported catalysts are highly active and selectivity as one of the main issues in CO₂ reduction is addressed. These points along with the fact that ferroelectric catalysis is a currently emerging hot topic, the manuscript is appropriate for Nature Communications and I recommend that the manuscript be published after the authors address the following comments:

1-1. Could the authors hypothesize how a practical device based on their catalyst would be constructed? How is the ferroelectricity switched? Does the increased efficiency outweigh the energetic cost for FE switching?

Response 1-1: Following the instructive suggestion of the reviewer, we propose a feasible device design where metal anchored α -In₂Se₃ monolayer is placed between the electrodes to achieve the ferroelectric switching and controllable catalysis, as shown in **Figure R1**. This device design is inspired by the recently fabricated ferroelectric diode from 2D α -In₂Se₃ layers (see **Nanoscale**, **10**, 14885, 2018). The polarization dependent electron transfer and *d* band center shift shown in this work allows the control of the reaction pathway and products of CO₂RR. We have added this device proposal and related discussions in the revised manuscript.

Figure R1 Schematic diagram of a feasible device where metal anchored α - In_2Se_3 monolayer is placed between the electrodes to achieve ferroelectric switching and controllable catalysis, tuned by the reversal of the bias direction.

For the last two questions, the DFT computations show that the overpotential for CO_2RR can be reduced by 0.39V, while the required potential to achieve ferroelectric switch is only 0.08V for the $\text{Pd}@\text{In}_2\text{Se}_3$ catalyst. This shows that the increased efficiency significantly outweighs the energy cost for the FE switching.

1-2. The authors always write "the ferroelectric switch". The term "ferroelectric switching" is more commonly used and I would suggest the authors to change all occurrences (starting with the title).

Response 1-2: we appreciate the reviewer's suggestion, following which we have replaced all "the ferroelectric switch" with "the ferroelectric switching".

1-3. Figure 1 would be clearer if cell boundaries would be shown (at least for the in-plane directions) such that the periodicity and viewing direction is more obvious. Also could the authors clarify if the charge density differences in Figure 1f are computed at the same geometry or if the density difference could be affected by atomic-relaxation artefacts? This is unfortunately not clear from the corresponding methods section in the SI.

Response 1-3: We thank the reviewer for this valuable suggestion to help improve the readability of our work. In response, we have added the cell boundaries in **Figure 1**. The charge density differences in **Figure 1f** are computed at the same geometry, the same are for

Figure S7. We have added the related descriptions in the figure captions.

1-4. I find the MD data shown in figure S6 quite weak as a support for the absence of structural changes. In that figure we merely see that the energy fluctuates ~ 0.05 eV but what these fluctuations correspond to remains completely unclear. The authors should show multiple configuration snapshots (at the very least two, one at the beginning and one at the end).

Response 1-4: We thank the reviewer for this suggestion, following which we have increased the simulation time up to 15 ps, and added five configuration snapshots, which correspond to the structures at each 3 ps MD simulation (see **Figure R2a**) (**Figure S10** in the *Supporting Information*). The structure of the catalyst (including the substrate) can stay stable at 300 K for at least 15 ps.

We also state in the figure caption that the energy fluctuation is from the thermal disturbance since the temperature effects have been considered during the AIMD simulations.

Figure R2 *Ab initio* molecular dynamics (AIMD) results of (a) the Pd@P↓-In₂Se₃ 2×2 super cell, where the energy fluctuation is from the thermal disturbance induced by the temperature; (b) 2Pd@P↓-In₂Se₃ unit cell for 15 ps with a time step of 1 fs at 300 K. The insert shows the

configuration snapshots of the initial state (IS) and the final state (FS).

1-5. In Figure 2 is the energy axis $E-E_{\text{Fermi}}$? If yes, this could be clarified in the figure. Also should it not be "sigma back donation"? It looks like sigma overlap to me.

Response 1-5: Yes, the energy axis is $E-E_{\text{F}}$, and it has been clarified in the figure. Also, the " π back donation" has been revised to " σ back donation" in the revised **Figure 2**.

1-6. When performing the analysis of the CO_2RR pathways, how were configurations screened? I.e. did the authors test multiple adsorption sites and adsorption modes? If yes, how many and how were they generated/selected?

Response 1-6: We have tested multiple adsorption sites and modes to determine the configurations of the intermediates. As shown in Figure S11, for each hydrogenation step, the sites of C and O atoms are considered as the possible H adsorption sites. On the condition that both configurations could be successfully optimized, we chose the one with lower Gibbs free energy. In the case that the configurations could not be successfully optimized, we adjust the relative locations of H atom and C/O atom, such as distance and angle, to make the configuration optimized. There are 167 configurations screened for the analysis of the CO_2RR pathways, and the related detailed data (POSCAR) are given in the Supplementary Data.

1-7. I find Figure 4a little confusing, in particular because different representations are used in a) and b). To be clearer, it would be nice if panel b adopted the same "to-the-left" and "to-the-right" scheme as panel a). This scheme could be further clarified by more clearly marking the starting point in the center.

Response 1-7: We greatly appreciate this suggestion by the reviewer, following which we have revised Fig. 4b accordingly (to make the starting point in the center). However, since CO_2 molecules cannot be activated on $\text{Zr@P}\downarrow\text{-In}_2\text{Se}_3$, the CO_2RR starting from the CO_2 hydronation will not take place, thus the first two steps on the left side are blank.

In contrast, CO_2 can be activated on $\text{Zr@P}\uparrow\text{-In}_2\text{Se}_3$, but the reaction is stuck at the step of forming the intermediate HOCHO^* . We show that the reaction can be reactivated provided that the polarization direction of In_2Se_3 is reversed from up to down, and the catalytic reduction can proceed without overcoming the huge barrier. Therefore, the reaction starts from HOCHO^* adsorbed $\text{Zr@P}\downarrow\text{-In}_2\text{Se}_3$ on the left side.

Figure R3 The free-energy profile for the CO₂ electrochemical reduction reactions along the minimum energy path at 0 V (vs. RHE) on (a) Rh@In₂Se₃, and (b) Zr@In₂Se₃. The insets show the optimized configurations of the intermediates. The pink (light blue) shaded area indicates the catalytic reaction when the polarization is pointing upwards (downwards).

1-8. The authors use PBE, would PBE+U for the TM d states change anything in the results? Also I believe there is a typo "gamma-pack" in the methods section that should be "gamma-centered Monkhorst-Pack".

Response 1-8: We thank the reviewer for the valuable comment on the U effect and for pointing out some typos.

To check the U effect on our main conclusions, we have conducted the PBE+U calculation with Pd@In₂Se₃ as a representative example to reexamine the reaction pathway, overpotentials, and final products of CO₂RR with the FE switching. Although the limiting potentials are slightly (less than 0.2 V) larger than the PBE values when the U is considered, all the main conclusions remain unchanged, see Fig. R4 (Fig. S20 in *Supporting Information*). The reaction path, final product, potential-limiting step, and the variation of limiting potential caused by polarization conversion are almost the same as the results from the PBE calculations. These findings show that the results from the standard PBE calculations provide

accurate predictions to describe the CO₂RR activity. To clarify this matter, we have added a section named “DFT vs. DFT+U” in the *Supporting Information*.

Figure R4 Comparison of the CO₂RR pathways on Pd@In₂Se₃ by using the PBE and PBE+U methods.

“The inclusion of the Hubbard-*U* term via, e.g., the DFT+*U* approach, may be more suitable for systems with highly localized orbitals. However, the DFT+*U* approach also suffers from a strong (linear) dependence of the energetics on the choice of the value of the parameter *U*, and on the choice of the localized projector functions that enter the definition of the *U*-dependent energy term. For example, the reduction energy (ΔH) of CeO₂→Ce₂O₃ process can vary between -5.1 ($U = 0$ eV) and -1.9 eV ($U = 5.0$ eV) using the DFT+*U* method, [*J. Phys. Chem. C* 112, 8643-8648, 2008] while the GGA-PBE value of -4.18 eV is in good agreement with the experimental measurements (-3.57 to -4.03 eV). On the other hand, the *U*-value is usually chosen based on its accuracy in reproducing the electronic structures (i.e., experimental band gap) of the bulk materials. However, to simulate catalysts, it is better to choose *U* to fit the energy of the oxidation-reduction, since catalytic processes are controlled by energy difference [*J. Phys. Chem. C* 115, 5841-5845, 2011]. The specific case in this work, namely CO₂RR on SAC surfaces, involves complex surface–adsorbate interactions, under which bulk-property derived *U* values in a locally changing surface environment may not adequately describe the reaction energetics. [*Phys. Chem. Chem. Phys.* 11, 9188-9199, 2009; *J. Phys. Chem. C* 121, 21343-21353, 2017].

Note that the results based on the GGA-PBE (the method used in this work) showed very good performance in understanding the reaction mechanisms and activity trends observed in experiments [*J. Am. Chem. Soc.* 141, 14115-14119, 2019; *Angew. Chem. Int. Ed. Engl.* 57, 16339-16342,

2018]. To verify the accuracy of the PBE results, we also investigated the CO₂RR pathways on the Pd@In₂Se₃ with the PBE+U method in which the previously validated *U* value of 8.00 eV was employed for the Pd 4*d* orbital. [*Phys. Rev. B*, 82, 184106, 2010]. Although the PBE+U results of limiting potentials are slightly (less than 0.2 V) larger than the PBE ones, the computed theoretical final product, reaction path, potential-limiting step as well as the variation of limiting potential caused by polarization conversion are the same (see **Figure R4**). Thus, the standard PBE calculations provided accurate predictions to describe the CO₂RR activity.”

The mentioned typos have been corrected, and we have carefully checked the whole manuscript.

1-9. While the article is written quite well, the text could benefit from thorough proofreading, ideally by a native speaker, as some conjugation mistakes, missing articles and other grammar issues exist. More particularly, the authors should try to use clearer formulations in places such as "does not have enough electrons" (page 2, top of right column), "high ratio of low-coordinates configurations" (page 2, middle of right column), "ensure the inert molecule chemically captured" (page 4, bottom of left column), "substract" should be substrate (page 5, top of left column), "depending on the ferroelectric surfaces" maybe use "ferroelectric state of the surface" (page 5, middle of right column), "the activations ... are much deeper" (page 7, top of left column).

Response 1-9: Following the reviewer comments, we have further polished the writing of the manuscript. In particular, all the mentioned issues have been corrected.

Reviewer #2 (Remarks to the Author):

This work reports a theoretical investigation of single-atom catalysts based on ferroelectric alpha-In₂Se₃ for CO₂ reduction. As the authors expected, alpha-In₂Se₃ may be applicable to CO₂ reduction. However, I still have some concerns about the stability of In₂Se₃ itself, which has to be ensured, particularly if the authors consider actual applications (see detailed comments below). Thus, I do not think that this work qualified enough to be published on Nature Communications.

Major concerns:

2-1. In this work, the authors mainly focused on how stable single metal atoms are on alpha-In₂Se₃. However, the more critical concern is the stability of the substrate itself.

Compounds consisting of In and Se are possible to have various compositions and structural polymorphs that can appear at typical atmospheric conditions like room temperature and atmospheric pressure. Thus, In-Se compounds have been studied for application to phase change memory. To use materials as electrochemical catalysts for CO₂R, the stability of the materials must be ensured especially under harsh environments like alkaline or acidic aqueous solutions. However, to the best of my knowledge, α -In₂Se₃ has not been proven to be stable and I am skeptical to its stability in light of the presence of various competing form of In-Se compounds. Furthermore, single atom catalysts need significant doping of α -In₂Se₃, but such a doping would deteriorate the phase stability of the material as well.

Response 2-1: We thank the reviewer for a careful assessment of our work, especially raising the critical comments and valuable suggestions for the improvement of the manuscript.

The reviewer's main concerns are with the stability of the ferroelectric substrate. We address all the raised issues below.

1. The intrinsic stability of α -In₂Se₃ and structural polymorphs

We agree with the reviewer that the In-Se compounds have many possible structural polymorphs, such as the phases of α , β , γ , δ , κ -In₂Se₃, which have been determined by X-ray diffraction and TEM (*Small*, 10, 2747, 2014). However, α -In₂Se₃ is the ground-state phase and is stable at the room temperature from both theoretical and experimental perspectives, as elaborated below.

Theory: To examine the stabilities of the different structural polymorphs, we have calculated the total energies of six possible phases of the In₂Se₃ monolayer, see **Figure R5**, including the β' , β , α and α' , zincblende and wurtzite phases. The α -In₂Se₃ monolayers with ferroelectric polarization have the lowest total energies, consistent with the recent theoretical work by Ding et al. [*Nat. Commun.* 8, 14956, 2017], indicating α -In₂Se₃ to be the most stable phase.

We note that there are two α -In₂Se₃ phases which share very similar atomic structures and are almost energetically degenerate, and further calculations show that they have the same CO₂RR performance as shown in **Figure R6**. We have added pertinent discussions into the revised manuscript and the figures into *Supporting Information*.

Figure R5 (a) Calculated total energy versus lattice constant for six In₂Se₃ monolayer phases. (b)-(g) Top and side views of these six In₂Se₃ monolayers, among which the structures shown in (b), (d), and (f) are derived from the fcc, wurtzite, and zinblend crystals, respectively.

Figure R6 Comparison of CO₂RR pathways on Pd@In₂Se₃ for the α and α' phases.

Experiments: Different phases of In₂Se₃ (α, β, γ, δ, κ-phase) have been experimentally synthesized (Small, 10, 2747, 2014), but under distinct fabrication conditions. Past reports have explicitly indicated that α-In₂Se₃ is the room-temperature phase (J. Appl. Crystallogr., 12, 416, 1979; J. Phys. Soc. Jpn., 21, 1848, 1966), while β, γ, δ -phases are high-temperature phases (J. Less Common. Met., 143, 83, 1988). Phase transformations can be achieved via the path of $\alpha \xrightarrow{200^\circ\text{C}} \beta \xrightarrow{520^\circ\text{C}} \gamma \xrightarrow{730^\circ\text{C}} \delta$ (Table 2 of Small, 10, 2747, 2014), which also indicates that

α phase is the stable room-temperature structure. This point has been further verified by the recent synthesis of α -In₂Se₃ layers that have been taken to fabricate ferroelectric devices [*Nano Lett.* 15, 6400-6405, **2015**; *Phys. Rev. Lett.* 120, 227601, **2018**; *Nat. Electron.* 2, 580, **2019**]. Cui *et al.* have pointed out that the cooling rate is critical for obtaining the α phase, which is stable at room temperature (*Nano Lett.* 18, 1253-1258, **2018**). Therefore, α -In₂Se₃ is rather stable at room temperature and can be well prepared by controlling the synthesizing temperature.

2. Stability of α -In₂Se₃ under harsh environments

We have not found experimental report regarding the stability of α -In₂Se₃ under alkaline or acidic aqueous solutions. However, we theoretically evaluated the electrochemical stabilities of α -In₂Se₃ monolayer by the dissolution potential U_{diss} , [*Electrochimica Acta*, 52, 5829-5836, **2007**; *J. Am. Chem. Soc.*, 142, 5709-5721, **2020**; *ACS Catal.*, 9, 11042-11054, **2019**], which is defined as

$$U_{diss} = U_{diss}^{\circ}(bulk) - E_f/ne$$

Where $U_{diss}^{\circ}(bulk)$ and n are the standard dissolution potential of In/Se bulk and the number of electrons involved in the dissolution, respectively, which can be obtained from the NIST database [N.C. WebBook, <https://webbook.nist.gov/chemistry/>]. E_f is the formation energy of In and Se atoms in the In₂Se₃ monolayer given by:

$$E_{f-se} = (E_{In_2Se_3} - 2E_{In} - 3E_{Se})/3$$

$$E_{f-In} = (E_{In_2Se_3} - 2E_{In} - 3E_{Se})/2$$

where E_{In} , E_{Se} are the respective total energies of the In and Se atoms in their most stable bulk structures, $E_{In_2Se_3}$ is the total energy of the In₂Se₃ monolayer. According to the definition, materials with $U_{diss} > 0V$ vs SHE are regarded as electrochemically stable under acidic conditions. The U_{diss} values of both In and Se in In₂Se₃ monolayer are positive (see Table S2), indicating the electrochemical stability of the In₂Se₃ monolayer.

3. Phase stabilities under metal doping

To study the stabilities after metal doping, we performed additional AIMD simulations with different doping concentrations (see **Figure R2a** and **2b**). The results indicate that α - In_2Se_3 monolayer with one Pd or two Pd adsorption on the surface are stable at room temperature (300 K) for 15 ps (no phase transformation, no metal clustering, no obvious energy decrease or increase, only showing fluctuations around a stable value due to thermal disturbance).

The above analysis shows that α - In_2Se_3 is a room-temperature stable phase based on extensive theoretical simulations and experimental demonstrations, and such stability can be preserved under the harsh environment and surface metal doping. We have added the related discussions into the revised manuscript.

2-2. As potential single-metal-atom sites, several atomic sites on the surface are accounted for (i.e., the dopants are assumed to act as interstitial defects) but introduced dopants may prefer to occupy other sites (e.g., substitution for In sites). To determine the most plausible atomic configurations of the single-metal-atoms, the thermodynamic defect formation energy should be checked, taking the chemical potentials of defect components into account.

Response 2-2: We thank the reviewer for raising this concern and making valuable suggestions. Below we provide detailed clarification and response.

For the interstitial adsorptions of metal atoms on the surfaces, all the possible sites have been investigated, and the most stable configurations have been used for the CO_2RR analysis.

For the possible substitution on the In sites mentioned by the reviewer, we have performed comprehensive calculations. The formation energy of the In vacancy and the diffusion barriers of an In atom removal from In_2Se_3 (Figure R7) have been calculated. The results show that the formation energies of the In vacancy are 1.33 and 1.52 eV for $\text{P}\uparrow$ - and $\text{P}\downarrow$ - In_2Se_3 , respectively. These high formation energies indicate the difficulty to form In vacancy from thermodynamic point of view. Moreover, the energy barriers for an In atom to diffuse from the subsurface to the top-surface (to create the vacancy) are 2.68 and 2.94 eV for $\text{P}\uparrow$ - and $\text{P}\downarrow$ - In_2Se_3 , respectively, which suggest a very small possibility to form In vacancy from kinetic point of view.

Figure R7. The calculated energy barriers for an In atom diffusion from the subsurface to the top-surface for $P\downarrow$ - and $P\uparrow$ - In_2Se_3 . The insets show the optimized structures of the initial states (IS), the transition states (TS), and final states (FS) of the In atom diffusion. The blue broken circle represents the initial position of the In atom.

Figure R8 Top and side views of the Pd-doped In_2Se_3 monolayer with (a) downward and (b) upward polarization ($P\downarrow$ and $P\uparrow$). (c) The free-energy profile for the first hydrogenation step of CO_2RR (COOH^*) on $\text{Pd}@/\text{In}_2\text{Se}_3$ and Pd-doped In_2Se_3 , respectively.

In addition, we find that the metal substituted In_2Se_3 is not suitable for the electrocatalytic CO_2RR , even if we do not consider the difficulties of the In vacancy formation and metal substitutions. According to the free-energy profile for CO_2 electrochemical reduction reactions along the minimum energy path at 0 V (vs. RHE) on Pd substituted $\text{P}\uparrow$ - and $\text{P}\downarrow$ - In_2Se_3 (namely Pd substitutes the In vacancy), the energy barriers of $\text{CO}_2 + * \rightarrow \text{OCOH}^*$ are up to 1.17 and 1.88 eV on Pd-doped $\text{P}\uparrow$ - and $\text{P}\downarrow$ - In_2Se_3 , respectively, due to the full occupations of the Pd-d orbitals (see **Figure R8**), leading to the low CO_2RR activities of these catalysts.

Overall, we conclude from our additional calculations and analysis that (i) In vacancies in In_2Se_3 are hard to form due to the high formation energies and diffusion barriers, making metal substituted In_2Se_3 rare in reality and (ii) metal substituted In_2Se_3 is unsuitable for CO_2RR . Therefore, we focus our discussions in this work on metal anchored In_2Se_3 on the surface.

2-3. When calculating binding energies of single metal atoms, which energy reference is considered for isolated metal atoms? For instance, Ref. 53 considered the energy per atom of the stable bulk metal phase when calculating the binding energies.

Response 2-3: We thank the reviewer for raising this question for clarification. In our binding energy computations, the energies of isolated TM atoms are taken as the reference due to the following considerations:

1) During the synthesis of SAC, the single metal atom normally is supplied by the mononuclear metal precursors instead of metal bulk [*Joule* 2, 1242–1264, **2018**]; for instance, $[(\text{NH}_3)_4\text{Pt}]^{2+}$ for Pt SAC [*J. Am. Chem. Soc.* 139, 14150–14165, **2017**] and $\text{Pd}(\text{hfac})_2$ for Pd SAC [*J. Am. Chem. Soc.* 137, 10484–10487, **2015**]. Dissociation of the metal bulk is not involved during the SAC fabrication.

2) We noticed that the energy per atom of bulk metal was used as the reference in some studies; however, most theoretical and experimental investigations used the energies of isolated TM atoms in order to reflect the real experimental process, see, e.g., *J. Am. Chem. Soc.* 139, 12480, **2017**; *J. Am. Chem. Soc.* 141, 14515, **2019** (at page 6 of *Supporting Information*); *Nature Commun.* 10, 5231, **2019**; *Nature Chem.* 3, 634, **2011**.

To provide further evidence that the metal single atoms on surface will not aggregate to form clusters, we not only showed the high migration barriers for each metal (**Figure S9**), but also

did additional calculations to obtain the clustering energy ($E_{cluster}$) of Pd@In₂Se₃ to estimate the clustering tendency of Pd atoms on the surface [*Int. J. Hydrogen Energ.* 37, 309-317, **2012**; *Phys. Rev. Lett.* 80, 3650-3653, **1998**; *J. Mater. Chem. A* 8, 20725-20731, **2020**]. The calculated values of $E_{cluster}$ are negative (-0.3 eV and -0.04 eV for Pd@P↓-In₂Se₃ and Pd@P↑-In₂Se₃, respectively), which means that the formation of the metal cluster is not energetically preferred. Besides, we have performed first-principles finite-temperature molecular dynamics simulations of two dispersedly adsorbed Pd atoms on P↓-In₂Se₃ (2Pd@P↓-In₂Se₃) with a Nose–Hoover thermostat at 300 K. The two Pd atoms maintain dispersedly adsorbed features for at least 15 picoseconds (see **Figure R2b**). All the above results confirm the high stability of the single-atom adsorption at room temperature, indicating that metal clustering is unlikely.

To clarify this matter, we have added a section with the title “**The clustering energy**” in the *Supporting Information*:

“The clustering tendency of Pd atoms on the surface is estimated by the clustering energy ($E_{cluster}$), [*Int. J. Hydrogen Energ.* 37, 309-317, **2012**; *Phys. Rev. Lett.* 80, 3650-3653, **1998**; *J. Mater. Chem. A* 8, 20725-20731, **2020**], which is defined as the difference in binding energies between a single metal atom ($E_{b,sin}$) and the metal dimer ($E_{b,dim}$):

$$E_{cluster} = E_{b,sin} - E_{b,dim}$$

where ($E_{b,sin}$) and ($E_{b,dim}$) are defined as:

$$E_{b,sin} = E_{Pd@In_2Se_3} - E_{In_2Se_3} - E_{Pd-bulk}$$

$$E_{b,dim} = \frac{1}{2}(E_{Pd_2/In_2Se_3} - E_{In_2Se_3} - 2E_{Pd-bulk})$$

where $E_{Pd-bulk}$ is the chemical potential of the Pd atoms in their bulk phase and E_{Pd_2/In_2Se_3} represents the total energy of the substrate with a Pd dimer. According to the definitions, negative values of $E_{cluster}$ mean that the metal cluster does not tend to form. The calculated values of $E_{cluster}$ are -0.3 eV and -0.04 eV for Pd@P↓-In₂Se₃ and Pd@P↑-In₂Se₃, respectively. To further analyze the stability of single-atom adsorption, we have performed first-principles finite-temperature molecular dynamics simulations of two dispersedly adsorbed Pd atoms on P↓-In₂Se₃ (2Pd@P↓-In₂Se₃) with a Nose–Hoover thermostat at 300 K. The fluctuations of the temperature and the total energy as a function of

the simulation time are given in **Figure R2b**. The two Pd atoms maintain dispersedly adsorbed features for at least 15 picoseconds. The distance between the two Pd atoms stays essentially unchanged. All these results confirm the dynamic stability of the single-atom adsorption at room temperature.”

2-4. 5-ps MD simulations at 300 K (I assume that the authors used NVT ensemble) may provide a meaningful insight into the stability of single metal atoms, but it does not give useful information for the overall stability of the catalyst including the support.

Response 2-4: To check the overall stability of the catalyst including the support, we performed AIMD simulations at 300K for up to 15 ps (longer time simulations are prohibitively expensive computationally). The structure of Pd@In₂Se₃, including the ferroelectric substrate, was not significantly deformed (see **Figure R2**). The total energies have the fluctuations around a stable value, but do not decrease or increase significantly during the 15 ps simulation time period. These data should be appropriate to indicate the overall stability of SAC.

2-5. The authors should mention how large supercells (i.e., single-metal-atom concentrations) are used. My feeling is that the dopant concentration considered in the present work is too large to occur in actual materials because the DOS of alpha-In₂Se₃, which is a semiconductor (PBE gap is around 0.4 eV?), is highly metallic when it is doped. Because the catalytic activity is likely to depend on the single-metal-atom concentration, the authors should first discuss the catalytic activity when the doping concentration is in dilute limit.

Response 2-5: Our SAC models are built based on the 2×2 α-In₂Se₃ supercell, which is now explicitly mentioned in the methodology part. To investigate the dependence of the catalytic activity on the metal atom concentrations, we have calculated the free-energy profile for CO₂ electrochemical reduction reactions along the minimum energy path at 0 V (vs. RHE) on Pd@In₂Se₃ by using 2×2, 4×4, and 6×6 α-In₂Se₃ supercells. As shown in **Figure R9**, the difference of the free energy profiles for CO₂RR between Pd@ In₂Se₃ (4×4) and Pd@In₂Se₃ (6×6) are within 0.05 eV, and these converged free-energy profiles indicate the catalytic activity when the doping concentration is in the dilute limit. On the other hand, though the binding strengths of the reaction intermediates for the dilute doping concentration are stronger than those for the high concentration doping case, the rate limited steps for CO₂RR are the same, and the over potential difference of CO₂RR between the high and low

concentrations is less than 0.13 eV. All these results confirm that the CO₂RR activities of metal doped In₂Se₃ with the high and low concentrations are quite similar, and the doping concentration on metal doped In₂Se₃ does not change the conclusion of our reported work.

We have added the CO₂RR performances on larger supercells into the *Supporting Information*, and the corresponding discussions into the revised manuscript.

Figure R9 The free-energy profile for the CO₂ electrochemical reduction reactions along the minimum energy path at 0 V (vs. RHE) on Pd@In₂Se₃ by using 2×2, 4×4, and 6×6 α-In₂Se₃ supercells.

2-6. In Figure S8, the energy difference between the down and up polarizations is only ~0.15 eV/UC. The value seems to me inconsistent with the binding energy difference of Pd depending on the polarization direction (-1.69 eV/atom vs. -1.14 eV/atom).

Response 2-6: We thank the reviewer for raising this concern. The binding energy difference between the Pd@P↓-In₂Se₃ and Pd@P↑-In₂Se₃ is 1.69 – 1.14 = 0.55 eV per Pd atom. Since the model is built based on a 2×2 α-In₂Se₃ supercell, containing four In₂Se₃ unit cells, the binding energy difference is 0.55 ÷ 4 ≈ 0.14 eV per In₂Se₃ unit cell (eV/UC). Here we convert the “eV/atom” into “eV/UC” to make direct comparison with the result of bare ferroelectric layer (0.13 eV/UC) [*Nat. Commun.* **8**, 14956, 2017]. We have clarified this in the revised manuscript.

2-7. I found that even if the DOS of the transition metal anchored on alpha-In₂Se₃ significantly varies depending on the polarization direction, the charge lost and CO₂ binding energy remains almost the same. Taking Pd@In₂Se₃ as an example, QTM = 0.14 and 0.12 and Eb-CO₂ = -0.84 and -0.79 eV for the down and up polarizations, respectively. This was

not clearly explained.

Response 2-7: We thank the reviewer for raising this concern and making valuable suggestions, following which we have provided detailed explanations in the revised manuscript, as elaborated below.

The DOS variations, electron transfer, and binding energy change have different mechanisms under the polarization switching. The DOS variation is directly related to the orbital shift under the polarization flip as revealed in see **Figure 1c**. In contrast, the charge transfer (as well as the CO₂ binding energy) is mainly related to the itinerant electrons induced by the polarization strength and direction. The lost charge and CO₂ binding energy is less sensitive to the ferroelectric switch.

2-8. Regarding Zr@In₂Se₃ catalysts, they mentioned that the ferroelectric switching can readily occur when HOCHO* exists. However, the calculation within periodic boundary condition enforces the ferroelectric switching that occurs across the entire lattice. In actual catalysts, single metal atoms are likely to be fairly apart from each other and such a ferroelectric switching is difficult to occur. Local ferroelectric switching around HOCHO* may occur, but in general, the local ferroelectric switching is energetically prohibitive.

Response 2-8: We thank the reviewer for these critical but valuable comments. We clarify these issues below.

To check if the local or whole ferroelectric switching is energetically preferred, we have built a supercell with HOCHO* adsorbed Zr@6×6 P↑-In₂Se₃ (**Figure R10a**) and calculated the energy differences of the structure with ferroelectric switching at only small (**Figure R10b**) or large (**Figure R10c**) local area around the Zr anchored site and the entire lattice (**Figure R10d**).

The calculated results indicate that the local ferroelectric switching is indeed energetically prohibitive, and the cases (b) & (c) have much higher energies than that of (d). However, the ferroelectric switching of the entire lattice is not difficult, even when single metal atoms are fairly apart from each other (24.41 Å): the energy difference is 0.4 eV/6×6cell or 0.045 eV/2×2cell, which is comparable to the data shown in the manuscript. Therefore, we conclude that the ferroelectric switching can readily occur throughout the entire lattice due to the energetic preference when the HOCHO* exists.

Figure R10 Calculated energies of $\text{Zr@In}_2\text{Se}_3(6 \times 6)$ with the HCOOH adsorption. Orange and purple areas stand for the up and down polarization areas, respectively. (a) The polarization of In_2Se_3 is pointing upwards; (b) polarization flip at the small area of Zr anchored site (25%); (c) polarization flip at the larger area of Zr anchored site (50%); (d) polarization flip at the entire lattice (100%).

2-9. In this work, PBE functional is used, but the free-energy diagram depends on a type of exchange-correlation functionals. Thus, the direct comparison of their results with other calculations is unfair. Moreover, RPBE or BEEF functionals are known to be more relevant for studying molecular adsorptions. They need to check if their conclusion changes when using such functionals.

Response 2-9: We thank the reviewer for the valuable suggestions, following which we have performed computations to check if our results will be affected by using different functionals, as detailed below.

To examine the possible dependence of onsite Coulomb interactions on the free-energy diagram, we have performed PBE+U calculations and found that our main conclusion stays unchanged, see more details in our response to comment 1-8 by the first reviewer. We also have recalculated the free-energy diagram of CO_2RR on $\text{Pd@In}_2\text{Se}_3$ with the RPBE

functional, and the results (computed theoretical final product, reaction path, potential-limiting step as well as the variation of limiting potential caused by polarization conversion) agree well with those obtained by PBE (see **Figure R11**). Therefore, the results using the PBE functional are robust. To clarify this matter, we have added a section titled “**PBE vs. RPBE**” in the *Supporting Information*.

Figure R11 Comparison of CO₂RR pathways on Pd@In₂Se₃ by using the PBE and RPBE methods.

Minor comments:

2-10. How did the authors calculate site-specific charge differences like QTM and QCO₂? Did they use Bader analysis?

Response 2-10: Yes, we used Bader analysis to calculate site-specific charge differences. We have clarified this in the “Method” section in the revised manuscript.

2-11. In Table S3, the entropies of liquid species are presented. How these values are obtained?

Response 2-11: These values are from WebBook, N. C., <https://webbook.nist.gov/chemistry/>. We have clarified this in the Table caption of Table S3 in the revised *Supporting Information*.

2-12. Besides the out-of-plane polarization, the direction of the in-plane polarization should be marked.

Response 2-12: We have marked the direction of the in-plane polarization in revised **Figure 1**.

Reviewer #3 (Remarks to the Author):

This paper presents a thorough first-principles study of single atom catalysts (SACs) for CO₂ electroreduction supported by ferroelectric In₂Se₃ semiconductor. This is certainly an important topic and the authors identify the optimal metal elements for In₂Se₃-supported SACs and also show how the change in ferroelectric polarization can be used to strongly modify the catalytic properties of these systems. However, to demonstrate that their results are of a broad interest, the authors must show or at least provide arguments for the superiority of the proposed SAC-In₂Se₃ catalysts to the state-of-the-art catalysts reported in recent work for one or more of the following: lower overpotential, greater selectivity, lower cost, ease of fabrication, etc. In other words, why should the community invest effort in trying to experimentally realize this system instead of using the more standard SAC substrates such as carbon?

Response: We thank the reviewer for the recognition of the novelty of our work. Following the reviewer comments, we have pointed out in the revised manuscript the superiority of ferroelectric catalysts and emphasized the superior selectivity and controllable reaction pathway & product via the polarization switching.

Technically, I think the authors must address the following issues.

1) The coupling of ferroelectricity and catalysis is certainly interesting. However, a monolayer In₂Se₃ with adsorbed metal atoms may have significantly different electronic structure, for example becoming metallic which would preclude the switching of the polarization by an applied electric field. By contrast, a thin layer of metal on top of a thick FE oxide material studied in previous work will still be an insulator in the bulk and therefore will show switching behavior. The authors must address this point and demonstrate that the adsorption of SAC does not destroy the ferroelectricity (that is the switchability of polarization with electric field) of the In₂Se monolayer system.

Response 3-1: We thank the reviewer for these comments.

We fully agree with the reviewer that the switchability of polarization is closely related with electronic structures since the ferroelectricity is generally from the offset positive/negative charge center, thus the semiconducting or insulating feature is essential to the ferroelectrics.

For the catalysts studied in this work, the metal anchored In₂Se₃ monolayer seems to be

metallized as shown in Figure 1 of our manuscript. However, such metallization is local, and the metallic states are mainly from the anchored metal atoms and localized in the immediate surrounding area. To prove this point, we have built a larger supercell with Pd anchored 6×6 $P \downarrow$ - In_2Se_3 monolayer as shown in the **Figure R12** (left). Based on the projected DOS analysis (see the right of **Figure R12**), obviously the metallic density of electronic states near the Fermi level is primarily from Pd and the area near the SAC adsorption site (in the red circle). The other parts (in the blue and green circles) away from the anchor site are semiconducting or insulating. The overall ferroelectricity of the metal anchored In_2Se_3 can thus be well preserved. Therefore, the system will show proper switching behaviors under the electric field.

For $\text{Pd}@ 6 \times 6 P \uparrow$ - In_2Se_3 , the semiconducting property is preserved since the d orbital of Pd is shifted by the polarization. The polarization switching behavior with electric field will remain. For other metal doped In_2Se_3 , the ferroelectricity and switching behaviors under electric field can be also preserved due to the same mechanism.

The above discussions and related figures are added to *Supporting Information* and the revised manuscript.

Figure R12 Partial density of states of the Pd SAC, which is constructed by one Pd atom on 6×6 (a) $P \downarrow$ - and (b) $P \uparrow$ - In_2Se_3 supercell. The red, green and blue lines represent the total

density of states of the selected areas circled by the red, green and blue lines, which are labeled as “near”, “middle” and “far” samples away from the Pd atom.

2) To demonstrate the stability of SAC with respect to agglomeration, the authors perform ab initio MD simulations at 300 K for 5 ps. This is not sufficient for exploring the potential energy surface and demonstrating that SAC will not diffuse because diffusion may take place on a time scale much longer than 5 ps. Therefore, a different approach must be used to demonstrate stability versus agglomeration (e.g. agglomeration reaction pathway calculations, or perhaps longer and/or higher temperature simulations).

Response 3-2: We thank the reviewer for the comments and suggestions.

Accordingly, we have recalculated the AIMD simulations with longer time (15 ps), as shown in the revised **Figure R2** (**Figure S10** in Supporting Information). During the 15 ps simulation period, we have not observed any phase transformation, metal clustering, or obvious energy decrease or increase (except fluctuation around a stable value due to thermal disturbance), thus it is reasonable to conclude that SAC is stable without metal diffusion.

Furthermore, we also have performed first-principles finite-temperature molecular dynamics simulations of two dispersedly adsorbed Pd atoms on P↓-In₂Se₃ (2Pd@P↓-In₂Se₃) with a Nose–Hoover thermostat at 300 K. The fluctuations of the temperature and the total energy as a function of the simulation time are given in **Figure R2**. The two Pd atoms maintain dispersedly adsorbed features for at least 15 picoseconds. The distance between the two Pd atoms stays essentially unchanged (see the inset of **Figure R2**). These results confirm the dynamic stability of the single-atom adsorption at room temperature.

Besides, the clustering energies also have been calculated to exclude the possibility of metal agglomerations, see more details in our response to comment 2-3 by the second reviewer.

3) For electrocatalysts, good conductivity of the electrode/catalyst is typically important. Since In₂Se₂ is a semiconductor, will this tend to decrease the electroreduction performance?

Response 3-3: We thank the reviewer for raising this important question. As mentioned in our manuscript, the pristine In₂Se₃ is an insulator, thus cannot activate electrochemical CO₂ reduction. Thus, we have introduced single TM atom to improve the local conductivity of the system, making the electrochemical CO₂ reduction possible.

Considering the above, while the authors do a good job proposing a new class of SAC CO₂

reduction catalysts, it is not clear from the present manuscript that this class is indeed promising and enables achievement of performance that is unavailable or even matches the performance obtained using other classes of SAC systems or that it provides a new mechanism for catalysis. If the authors make a convincing case for the promise of these catalysts the paper may be suitable for publication in Nature Communications.

Response: we thank the reviewer again for the positive assessment of our reported work. Following the reviewer comments/suggestions, we have performed additional comprehensive calculations and analysis to demonstrate the stability of the catalysts and verify the feasibility to achieve controllable CO₂RR *via* the ferroelectric switching by different methodologies and different models. Moreover, we also show that the polarization flip is not significantly affected by the surface metal adsorption.

We hope that our clarifications and revisions have well addressed all the concerns of the three reviewers. Our study unveils unique catalytic performance of transition metals anchored on the In₂Se₃ monolayer that is adjustable *via* the ferroelectric switching, which is absent in other SACs, and introduces a new catalytic mechanism (polarization shifted *d* band centre and electron transfer) for controllable catalysis. On this basis, we think that this work will open a new avenue for 2D ferroelectric catalysis and thus meet the criteria of *Nature Communications*.

REVIEWER COMMENTS

Reviewer #1 (Remarks to the Author):

The authors have addressed all my previous concerns in a satisfactory fashion and have improved the manuscript considerably.

Reviewer #2 (Remarks to the Author):

Authors have addressed many concerns raised by the reviewers, but more clarifications are needed to recommend for publication.

1. I did not see how many dopants can be incorporated in interstitial sites. It is better to explicitly calculate the formation energy of interstitial dopants in order to show SAC based on In₂Se₃ can actually have a large number of active sites.
2. Taking the energy of an isolated atom as a reference in calculations of binding energy does not reflect experiments wherein molecular precursors are used to dope systems. An isolated metal atom is definitely less stable than the metal in a molecule. For the binding energy to have proper physical meaning, the authors should set an adequate energy reference. Otherwise, we cannot judge the binding strength based on the binding energy. In addition, using an isolated atom as a reference is likely to lead to fairly large binding energies of TMs because of instability of an isolated atom.

Reviewer #3 (Remarks to the Author):

The authors have answered most of the criticisms from the previous review round. However, two issues are still unresolved. The more important is the possible agglomeration of the Pd atoms into nanoparticles. The authors have now performed 15 ps AIMD simulations to demonstrate that agglomeration is not favorable. These simulations are only marginally more useful than the 5 ps AIMD simulations performed previously because the agglomeration may occur on the ns or even microsecond timescale. So extending the simulation timescale from 5 ps to 15 ps is insufficient. I understand that AIMD simulations on ns timescale are impossible. Therefore, a different method must be used to demonstrate the lack of agglomeration. The authors take a step towards such a demonstration in their calculations of the dimer energy versus the single-atom Pd adsorption. However, even though the dimer is higher in energy than single atom, this does not yet prove that agglomeration is thermodynamically unfavorable. For example, while a dimer may be unfavorable, a particle where most of Pt atoms are bound to other Pt atoms rather than In₂Se₃ surface (i.e. a 3D Pd particle) may be favorable so that the dimer state would represent a possible barrier state but does not fully describe the thermodynamic potential surface of this system.

For a complete description of this problem, a kinetic Monte Carlo simulations would be a good method. I am not sure if this is required in this case, but something more sophisticated than dimer energy calculation and 15 ps AIMD is certainly necessary.

Second, with regard to the issue of stability of alpha-In₂Se₃ under electrochemical environment and doping, it is not enough to consider the small perturbation of the structure under doping because the structure may be trapped in the alpha-In₂Se₃ local minimum, whereas in experiment other phases may be the global energy minimum in the presence of doping and electrochemical potential and possible effects of H⁺ and counterions. Thus, to demonstrate that alpha-In₂Se₃ is the global minimum energy phase even in the presence of dopants and in electrochemical environment, the other possible

In/Se phases with dopants and in the presence of H, OH and possibly counterions should be compared to α -In₂Se₃ with dopants and in the presence of H, OH and possibly counterions. I think this point is less crucial than the question of agglomeration because even if another phase is the global minimum in the presence of dopants, the α -In₂Se₃ may still be kinetically trapped and stable. Nevertheless, this still should be addressed

REVIEWERS' COMMENTS

Reviewer #3 (Remarks to the Author):

The authors have successfully addressed my concerns regarding their evaluation of the stability of the SAC and the alpha-In₂Se₃ phase. The paper is now suitable for publication in Nature Communications.

Responses to Reviewers' Comments

Changes made in the revised manuscript (the revised contents have been highlighted in red):

1. To indicate the stabilities of the ferroelectric SACs under the metal adsorptions and H*/OH*, the sentence “**even in the presence of adatoms and in electrochemical environment (see Figure S2 and Table S1-S2)**” has been added in paragraph 3 on page 2.
2. The sentence “**To provide the reference for the binding strengths, we have compared the binding energies of TM atoms in ferroelectric SACs with the ones in the corresponding molecular precursors, and it is found that the adsorption of TM (e.g. Pd, Rh and Ni) atoms on In₂Se₃ is energetically preferred, indicating the feasibility of the SACs to be synthesized based on the corresponding molecular precursors (see Table S4)**” has been added in paragraph 2 on page 3, in response to question 2 from Reviewer 2.
3. The sentence “**Possible metal (e.g., Pd and Nb) agglomerations are excluded by kinetic Monte Carlo (KMC) simulations, which show no metal-cluster formation on the surface at the room-temperature over a 100-second simulation period (see Figure S12-S14).**” has been added in paragraph 1 at page 4, to address the critical question about the possible agglomerations.
4. The discussion of “**since it corresponds to the coverage of 25% with the optimal adsorption energies, as shown in Figure S23. Four TM atoms uniformly distributed on the hexagonal centres of 4×4 In₂Se₃ supercell act as the catalytic active sites.**” has been added in Method on page 8, to indicate the catalytic sites on the surfaces.

Changes made in the revised Supporting Information (the revised contents have been highlighted in red):

1. Discussions about total energy of possible In/Se phases in the presence of metal adatoms and in the electrochemical environment have been added on page S3-S4. An additional figure (Figure S2) and table (Table S1) are also inserted.
2. Discussions about the energy difference between the binding energy of TM single atom in molecular precursors and in SACs have been added on page S12-S13. Table S4 is inserted.
3. Discussions about the kinetic Monte Carlo simulations have been added on page S16-S20. Table S5 and Figures S12-14 are inserted.
4. Discussions about the number of catalytic active sites have been added on page S42. Figure S23 is inserted.
5. The order of figures and tables has been updated accordingly.

Reviewer #1 (Remarks to the Author):

The authors have addressed all my previous concerns in a satisfactory fashion and have improved the manuscript considerably.

Response: We appreciate the Reviewer's positive evaluation.

Reviewer #2 (Remarks to the Author):

Authors have addressed many concerns raised by the reviewers, but more clarifications are needed to recommend for publication.

2-1. I did not see how many dopants can be incorporated in interstitial sites. It is better to explicitly calculate the formation energy of interstitial dopants in order to show SAC based on In_2Se_3 can actually have a large number of active sites.

Response 2-1: We thank the Reviewer for her/his comment and valuable suggestion, following which we have performed additional simulations to study the maximum active sites of Pd adatoms on a 4×4 In_2Se_3 supercell. Since the centers of the six-membered ring have been identified as the energetically preferred dopant sites (**Figure 1**), the metal atoms are uniformly dispersed (rather than forming clusters) as the catalytic active sites. Based on these results, we calculated the formation energies to determine the maximum active sites of SAC, with Pd adatoms evenly distributed in the 4×4 In_2Se_3 supercell as a representative. As shown in **Figure R1**, for both $\text{Pd}@P\uparrow\text{-In}_2\text{Se}_3$ and $\text{Pd}@P\downarrow\text{-In}_2\text{Se}_3$, when the surface converge is within 25% (i.e., 4 adatoms are uniformly distributed on the supercell surface), the formation energies increase (in absolute values) with increasing number of TM atoms, and converge to -1.69 and -1.15 eV/Pd, respectively. Additional Pd atoms on the surface will lower the (absolute) value of the formation energy due to the reduced nearest-neighbor distance and the resulting Coulomb repulsion. For example, the fifth Pd adatom on the 4×4 In_2Se_3 leads to the formation energies of -1.65 and -1.12 eV/Pd atom for the two polarization states. Therefore, we conclude that In_2Se_3 surface can host a large number of catalytic active sites, which are uniformly dispersed with the converge up to 25%.

The related discussions have been added into the revised manuscript, and the new results are added into the *Supporting Information*.

Figure R1 Calculated formation energy versus adatom number (n) for $n\text{Pd}@In_2Se_3 (4\times 4)$.

2-2. Taking the energy of an isolated atom as a reference in calculations of binding energy does not reflect experiments wherein molecular precursors are used to dope systems. An isolated metal atom is definitely less stable than the metal in a molecule. For the binding energy to have proper physical meaning, the authors should set an adequate energy reference. Otherwise, we cannot judge the binding strength based on the binding energy. In addition, using an isolated atom as a reference is likely to lead to fairly large binding energies of TMs because of instability of an isolated atom.

Response 2-2: We are grateful to the Reviewer for this highly helpful suggestion. In response, to facilitate the evaluation of the binding strength, we took the binding energies of TM atom in the molecular precursors of the SACs as the reference. For Pd-based SAC, the energy difference $E_{\Delta b}$ between the binding energy of Pd atom in $Pd(\text{hfac})_2$ (molecular precursor for Pd SAC [*J. Am. Chem. Soc.* 137, 10484–10487, 2015]) ($E_{b,Pd(\text{hfac})_2}$) and that in $Pd@In_2Se_3$ (E_{b-Pd}) is calculated to reflect the binding strength of Pd atom in $Pd@In_2Se_3$.

$$E_{\Delta b} = E_{b-Pd} - E_{b,Pd(\text{hfac})_2}$$

The binding energy of Pd atom in $Pd(\text{hfac})_2$ ($E_{b,Pd(\text{hfac})_2}$) is defined as:

$$E_{b,Pd(\text{hfac})_2} = E_{Pd(\text{hfac})_2} - 2E_{\text{hfac}^*} - E_{Pd}$$

where $E_{Pd(\text{hfac})_2}$ and E_{Pd} are the total energies of $Pd(\text{hfac})_2$ and isolated Pd atom, respectively.

E_{hfac^*} is defined as:

$$E_{\text{hfac}^*} = E_{\text{HfacH}} - \frac{1}{2}E_{\text{H}_2}$$

where E_{HfacH} and E_{H_2} are the total energies of the hexafluoroacetylacetonate and hydrogen molecules, respectively.

The calculated values of $E_{\Delta b}$ are -0.70 and -0.15 eV for Pd@P↓-In₂Se₃ and Pd@P↑-In₂Se₃, respectively. We can thus conclude that: 1) the adsorption of Pd atom on In₂Se₃ is energetically more favorable than that of Pd(hfac)₂; 2) it is highly feasible to synthesize Pd@In₂Se₃ by using Pd(hfac)₂ as the molecular precursor.

Using the same methods, we also calculated the energy differences $E_{\Delta b}$ for Rh- and Ni-based catalysts (see **Table R1**). All the calculated $E_{\Delta b}$ values are negative, indicating the good stability of our predicted SACs and the high promise to synthesize them starting from these molecular precursors.

For Re-, Zr- and Nb-based SACs, we did not find the proper corresponding molecular precursors in the literature. However, we noticed that the metal rods were used as the sources for the fabrications of single atom catalysts [*Nat. Commun.*, 4, 1924, **2013**]. Therefore, we kept the binding energies of these SACs calculated from isolated atoms in our work, which can be used to reflect the binding strengths.

Table R1 Calculated binding energies (in eV) of TM atoms in their molecular precursors ($E_{b,\text{precursor}}$) and SACs ($E_{b,\text{SAC}}$), as well as the energy difference between $E_{b,\text{precursor}}$ and $E_{b,\text{SAC}}$ ($E_{\Delta b}$).

Precursor	$E_{b,\text{precursor}}$	SAC	$E_{b,\text{SAC}}$	$E_{\Delta b}$
^a Pd(hfac) ₂	-0.99	Pd@P↓-In ₂ Se ₃	-1.69	-0.70
		Pd@P↑-In ₂ Se ₃	-1.14	-0.15
^b Rh(acac) ₃	-0.01	Rh@P↓-In ₂ Se ₃	-4.09	-4.08
		Rh@P↑-In ₂ Se ₃	-2.36	-2.35
^c Ni(acac) ₂	-1.42	Ni@P↓-In ₂ Se ₃	-2.72	-1.30
		Ni@P↑-In ₂ Se ₃	-1.97	-0.55

^a *J. Am. Chem. Soc.* 137, 10484, **2015**.

^b *Nat. Nanotechnol.* 15, 390–397, **2020**.

^c *ACS Appl. Energy Mater.* 1, 5286, **2018**.

We have added the discussions into the revised manuscript, and the table and detailed results into the *Supporting Information*.

Reviewer #3 (Remarks to the Author):

Authors have addressed many concerns raised by the reviewers, but more clarifications are needed to recommend for publication.

3-1. The authors have answered most of the criticisms from the previous review round. However, two issues are still unresolved. The more important is the possible agglomeration of the Pd atoms into nanoparticles. The authors have now performed 15 ps AIMD simulations to demonstrate that agglomeration is not favorable. These simulations are only marginally more useful than the 5 ps AIMD simulations performed previously because the agglomeration may occur on the ns or even microsecond timescale. So extending the simulation timescale from 5 ps to 15 ps is insufficient. I understand that AIMD simulations on ns timescale are impossible. Therefore, a different method must be used to demonstrate the lack of agglomeration. The authors take a step towards such a demonstration in their calculations of the dimer energy versus the single-atom Pd adsorption. However, even though the dimer is higher in energy than single atom, this does not yet prove that agglomeration is thermodynamically unfavorable. For example, while a dimer may be unfavorable, a particle where most of Pt atoms are bound to other Pt atoms rather than In₂Se₃ surface (i.e. a 3D Pd particle) may be favorable so that the dimer state would represent a possible barrier state but does not fully describe the thermodynamic potential surface of this system. For a complete description of this problem, a kinetic Monte Carlo simulations would be a good method. I am not sure if this is required in this case, but something more sophisticated than dimer energy calculation and 15 ps AIMD is certainly necessary.

Response 3-1: We thank the Reviewer for raising these concerns, especially for her/his instructive suggestions, following which we have performed kinetic Monte Carlo (kMC) simulations with the time scale up to 100 seconds to check for possible agglomerations at different annealing temperatures, with TM@In₂Se₃ (TM=Pd and Nb) as the representative example.

KMC simulation method. The details of KMC method have been described in previous works. [*Appl. Phys. Lett.* 88, 263116, **2006**; *Phys. Rev. B* 73, 195322, **2006**; *Phys. Rev. B* 77, 245322, **2008**.] Here, we start with a perfect α -In₂Se₃ monolayer with evenly distributed single metal atoms. The simulation size of the α -In₂Se₃ monolayer is $\sim 400 \times 400 \text{ \AA}^2$ with the periodic boundary conditions and the density of Pd monomers is $\sim 4\%$ (the same model and method also used for Nb case). During the annealing process, all Pd atoms (including Pd monomers and Pd atoms in Pd_N (N \leq 3) clusters) can diffuse to the nearest-neighbour sites on α -In₂Se₃ monolayer. For simplicity, we ignore the diffusion

of Pd atoms in Pd_N (N>3) clusters because large Pd_N (N>3) clusters are very stable. Three elementary rate processes are emphasized in our KMC model: (1) Diffusion of Pd monomers (D1); (2) Diffusion of Pd atoms in Pd₂ clusters (D2); and (3) Diffusion of Pd atoms in Pd₃ clusters (D3). We denote the activation barriers of the three processes by V_{D1} , V_{D2} , and V_{D3} , respectively, and the corresponding rates by R_{D1} , R_{D2} , and R_{D3} , with $R_i = v_0 \exp(-V_i/k_B T)$, where V_i is the activation barrier for process i , k_B is the Boltzmann's constant, and T is the annealing temperature. The attempt frequency is chosen as $v_0 = 2k_B T/h = 4.167 \times 10^{10} T$, in which h is the Planck's constant. All the values of the activation barriers used in our KMC simulations are from the DFT calculations, which are listed in **Table R2**.

Table R2 Values of activation barriers used in the KMC simulations obtained from DFT calculations: binding energies of metal monomers and metal atoms in metal clusters (E_i in eV/atom), and diffusion barriers of metal monomers and metal atoms in metal clusters (V_i in eV/atom).

		E_i	V_i
Pd@P↓-In₂Se₃	Pd monomer	-1.69	1.19
	Pd atoms in Pd ₂ cluster	-1.38	0.88
	Pd atoms in Pd ₃ cluster	-1.99	1.49
	Pd atoms in Pd ₄ cluster	-2.66	2.16
Pd@P↑-In₂Se₃	Pd monomer	-1.14	0.84
	Pd atoms in Pd ₂ cluster	-1.10	0.80
	Pd atoms in Pd ₃ cluster	-1.52	1.22
	Pd atoms in Pd ₄ cluster	-1.93	1.63
Nb@P↓-In₂Se₃	Nb monomer	-7.32	4.63
	Nb atoms in Pd ₂ cluster	-6.48	3.79
	Nb atoms in Pd ₃ cluster	-6.94	4.25
	Nb atoms in Pd ₄ cluster	-7.46	4.77
Nb@P↑-In₂Se₃	Nb monomer	-5.92	4.21
	Nb atoms in Pd ₂ cluster	-5.82	4.11
	Nb atoms in Pd ₃ cluster	-5.86	4.15
	Nb atoms in Pd ₄ cluster	-6.37	4.66

KMC simulation results. The P↓-In₂Se₃ monolayer with evenly distributed Pd atoms is chosen as the initial configuration to simulate the annealing process (**Figure R2a**). **Figure R2b-2f** show the surface morphologies of Pd@P↓-In₂Se₃ after 100 s annealing process at different temperatures. At

lower temperatures of $T \leq 350$ K (**Figure R2b**), the diffusion of Pd monomers does not happen, and after annealing the surface morphology of the Pd@P \downarrow -In₂Se₃ stays the same as in the initial case. At intermediate temperatures of $T = 390 \sim 410$ K (**Figure R2c and 2d**), the Pd monomers start to diffuse on α -In₂Se₃ monolayer, but no Pd_N ($N \geq 2$) cluster is formed during the annealing process. When temperature is further increased to $T = 470$ K (**Figure R2e**), the Pd_N ($N \geq 4$) clusters start to form, but the majority of Pd adatoms are monomers, indicating that single Pd atoms on the α -In₂Se₃ monolayer are not stable above 470 K. At the higher temperature of $T = 490$ K (**Figure R2f**), almost all the Pd adatoms form Pd_N ($N \geq 4$) clusters. Considering that the Pd@P \downarrow -In₂Se₃ are stable for $T \leq 410$ K, our kMC simulations clearly show that Pd@P \downarrow -In₂Se₃ electrocatalyst has a high stability in real electrocatalytic reaction conditions without agglomerating at room temperature (i. e., 293K).

The kMC results for Pd@P \uparrow -In₂Se₃ are similar (**Figure R3**), though the starting temperature to form the Pd cluster is much lower than that of Pd@P \downarrow -In₂Se₃ due to lower diffusion barrier (see **Table R2**), and no Pd clusters form at $T \leq 300$ K for 100 seconds, indicating the stability at room temperature. Since Pd@P \uparrow -In₂Se₃ has the lowest migration barrier of 0.84 eV (see Table 1 in the manuscript), other ferroelectric SACs are not expected to have the TM clustering issue at room temperature. For example, the metal agglomerations on ferroelectric surfaces of Nb@P \uparrow -In₂Se₃ and Nb@P \downarrow -In₂Se₃ do not occur even at 600K for 100s (see **Figure R4**).

Based on these kMC simulations together with our DFT findings, we are confident that the SACs are stable and will not agglomerate at moderate reaction environments.

$\text{Pd}@P\downarrow\text{-In}_2\text{Se}_3$

Figure R2 (a) The initial surface morphology of $\alpha\text{-In}_2\text{Se}_3$ monolayer in downward ($P\downarrow$) polarization with evenly distributed Pd atoms ($\text{Pd}@P\downarrow\text{-In}_2\text{Se}_3$). The surface morphologies of the $\text{Pd}@P\downarrow\text{-In}_2\text{Se}_3$ electrocatalysts annealed for 100 s at different temperatures: (b) 350 K, (c) 390 K, (d) 410 K, (e) 470 K, and (f) 490 K. Note that no Pd clusters are formed at $T \leq 410$ K.

Pd@P \uparrow -In₂Se₃

Figure R3 (a) The initial surface morphology of α -In₂Se₃ monolayer in upward (P \uparrow) polarization with evenly distributed Pd atoms (Pd@P \uparrow -In₂Se₃). The surface morphologies of the Pd@P \uparrow -In₂Se₃ electrocatalysts annealed for 100 s at different temperatures: (b) 290 K, (c) 310 K, (d) 330 K, (e) 350 K, and (f) 370 K. No Pd clusters are formed at T \leq 300K.

Figure R4 The initial surface morphology of α -In₂Se₃ monolayer with evenly distributed Nb atoms ((a) Nb@P↓-In₂Se₃; (d) Nb@P↑-In₂Se₃). Their surface morphologies of the electrocatalysts annealed for 100 s at different temperatures: (b), (e) 400 K; (c), (f) 600 K. Note that no Nb clusters are formed at $T \leq 600$ K

We have added the related discussions and results into the revised manuscript and *Supporting Information*.

3-2. Second, with regard to the issue of stability of α -In₂Se₃ under electrochemical environment and doping, it is not enough to consider the small perturbation of the structure under doping because the structure may be trapped in the α -In₂Se₃ local minimum, whereas in experiment other phases may be the global energy minimum in the presence of doping and electrochemical potential and

possible effects of H⁺ and counterions. Thus, to demonstrate that α -In₂Se₃ is the global minimum energy phase even in the presence of dopants and in electrochemical environment, the other possible In/Se phases with dopants and in the presence of H, OH and possibly counterions should be compared to α -In₂Se₃ with dopants and in the presence of H, OH and possibly counterions. I think this point is less crucial than the question of agglomeration because even if another phase is the global minimum in the presence of dopants, the α -In₂Se₃ may still be kinetically trapped and stable. Nevertheless, this still should be addressed.

Response 3-2: We thank the Reviewer for raising this concern and her/his valuable suggestion, which greatly helped us improve this work.

Accordingly, we examined if α -In₂Se₃ is still the global minimum energy phase in the presence of metal adatoms and in the electrochemical environment by calculating the total energies of other four phases (β' , β , wurzite, and zinblende) with presence of Pd, H* or OH*. As shown in **Figure R5**, for all these cases, the α -In₂Se₃ has the lowest total energy among five phases.

Figure R5 Calculated total energies for α , β' , β , wurzite, and zinblende phases of Pd@In₂Se₃ and H* and OH* adsorbed Pd@In₂Se₃. Structural configurations of OH*-Pd@In₂Se₃ are shown as insets while similar results for H*-Pd@In₂Se₃ and Pd@In₂Se₃ are not shown here.

For other phases with different In/Se ratios, we only considered the InSe monolayer since it has been experimentally synthesized [*Nat. Nanotechnol.* 12, 223, **2017**; *Adv. Mater.* 25, 5714, **2013**]. Due to

the different species and number of atoms, we cannot simply use the total energies to make a comparison between Pd@InSe and Pd@ α -In₂Se₃. We use the cohesive energy instead.

The cohesive energy of Pd@InSe ($E_{f,\text{Pd@InSe}}$) is defined as follows:

$$E_{f,\text{Pd@InSe}} = (E_{\text{Pd@InSe}} - n_1 E_{\text{In-bulk}} - n_2 E_{\text{Se-bulk}} - E_{\text{Pd-bulk}}) / N$$

where $E_{\text{In-bulk}}$, $E_{\text{Se-bulk}}$ and $E_{\text{Pd-bulk}}$ are the energies of the In, Se and Pd atoms in their most stable bulk structures, respectively; n_1 , and n_2 are the numbers of the In and Se atoms in Pd@InSe; N is the total number of atoms in Pd@InSe. The cohesive energies of Pd@In₂Se₃ ($E_{f,\text{Pd@In}_2\text{Se}_3}$), Pd@InSe ($E_{f,\text{H}^*-\text{Pd@InSe}}$, $E_{f,\text{OH}^*-\text{Pd@InSe}}$) and Pd@ α -In₂Se₃ with H* or OH* adsorptions ($E_{f,\text{H}^*-\text{Pd@In}_2\text{Se}_3}$, $E_{f,\text{OH}^*-\text{Pd@In}_2\text{Se}_3}$) are similarly defined and calculated. From the cohesive energies listed in **Table R3**, it is clear that α -In₂Se₃ always has the lower formation energy than InSe, for both pristine phase and that with H* or OH* adsorption.

All the above results indicate that α -In₂Se₃ is the global minimum energy phase, even in the presence of adatoms and in electrochemical environment.

Table R3 Calculated cohesive energy (in eV/atom) of pristine Pd@InSe and α phase of Pd@In₂Se₃, as well as those with H* or OH* adsorption.

		H*-adsorbed	OH*-adsorbed
Pd@InSe	-0.456	-0.358	-0.324
Pd@In ₂ Se ₃	-0.530	-0.452	-0.445

The related discussions have been added in the revised manuscript, while the new results are added into the *Supporting Information*.